# Testing the Corrosion Rate of Prestressed Concrete Beams Under Variable Temperature and Humidity Conditions

**DOI:** 10.3390/ma18071553

**Published:** 2025-03-29

**Authors:** Zofia Szweda, Michał Krak, Szymon Czerniak, Artur Skórkowski, Jacek Małek, Jakub Sikorski, Jakub Trojan, Petr Konečný, Miroslav Vacek, Jakub Marek

**Affiliations:** 1Department of Building Structures, Faculty of Civil Engineering, Silesian University of Technology, 44-100 Gliwice, Poland; mk307668@student.polsl.pl (M.K.);; 2Department of Measurement Science, Electronics and Control, Faculty of Electrical Engineering, Silesian University of Technology, 44-100 Gliwice, Poland; artur.skorkowski@polsl.pl (A.S.);; 3Department of Structural Mechanics, Faculty of Civil Engineering, VSB-Technical University of Ostrava, 70800 Ostrava, Czech Republicmiroslav.vacek@vsb.cz (M.V.);

**Keywords:** corrosion rate, prestressed concrete beams, temperature rise, humidity rise, scratching, long-term loading, strength of beams, LPR method, chloride migration, digital image correlation

## Abstract

To date, many studies can be found in the literature attempting to explain the effects of temperature and humidity on the rate of corrosion processes. However, it is difficult to analyze the results of these studies and draw unambiguous conclusions due to the different test conditions as well as different electrochemical test methods for corrosion rates. Most of these studies concern concrete reinforced with ordinary steel. However, there is a lack of research and analysis conducted on prestressed elements. The purpose of this study was to evaluate the effect of temperature and humidity changes on the development of corrosion processes in prestressed concrete beams. Tests were performed both under conditions of increasing temperature and humidity, which were reproduced in a climatic chamber, as well as in an environment exposed to chloride ions. The process of migration of chloride ions into the concrete was accelerated by the application of an electric field. In addition, selected beams were subjected to prolonged loading to sustain the induced scratching. Corrosion rate tests were carried out using the non-destructive linear polarization method (LPR). Strength tests of the beams were also carried out, as well as displacement and deformation measurements using the Aramis system’s digital image correlation technique. The beams without chloride addition had a fairly stable low level of corrosion current density throughout the test period, indicating the passive state of the reinforcement, regardless of the environment in which they were placed and the additional loading. In an environment with a humidity of 30% and a temperature of 20 °C, the corrosion current density increment was much faster than for beams with chloride additives in an environment with a humidity of 90% and a temperature of 30 °C. A smaller increase in corrosion current density could be observed in beams that were scratched, compared to non-scratched beams. The results of the strength tests indicated that in beams subjected to accelerated migration of chloride ions, the deflection at scratching was significantly lower than in beams without chloride addition. Also in these beams, milder strains were registered on the surface of the elements at the time of scratching.

## 1. Introduction

In the modern world, the realization of increasingly bolder structures in terms of height or span is being pursued. This is possible mainly thanks to the use of prestressed constructions, in which a system of forces that induces a state of stress opposite to that arising during the life of the structure has been deliberately introduced [1]. Failures of prestressed structures are caused primarily by failure to meet the requirements of corrosion protection, which can eventually lead to a structural failure. This is due to the susceptibility of prestressing steel to stress corrosion scratching, which manifests itself by rapid rupture in the elastic range, as well as sensitivity to local damage [2]. Pitting corrosion caused by chloride ions changes the adhesion of tendons to concrete and reduces the active cross-section of the tendon. In prestressed concrete structures, the adhesion of tendons to concrete is of fundamental importance [3].

The mechanical properties of tendons in prestressed structures are significantly affected by corrosion processes. This results in a reduction in the strength of the strands, as well as deformations at which the tendons break [4]. High-strength prestressing steel is more sensitive to corrosion processes compared to reinforcing steel, as indicated by the results of the tensile test-fatigue test-stress corrosion test. They show that at a corrosion loss of 6.3%, the strength of the strands decreases by about 20%, with an almost 5-fold reduction in the limiting deformation, while at a corrosion of 20.4%, the strength is only 38% of the original value, with less reduction in deformation at rupture [5]. On steel strings, the seeds of corrosion processes can occur during transport to the site of incorporation or during improper storage, where they are exposed to moisture along with other environmental factors such as Cl ions. As a result, pitting corrosion occurs, which can result in damage to the tendons when either tensioning or using the structure.

Since the prestressing tendons are made of high-strength steel, the yield strength of which is two to three times greater than that of ordinary reinforcement, the loss of cross-sectional area of the prestressing tendons will result in a failure that is two to three times worse than that of ordinary reinforcement. In addition, prestressing tendons are subject to other failure mechanisms, such as hydrogen embrittlement and hydrogen-induced stress corrosion cracking. Atomic hydrogen can be formed as part of the corrosion process and penetrate the steel in corrosion pits in environments with a pH below 7. Under such conditions, the ductility of the prestressing tendons is suddenly reduced, and brittle fractures occur at low strain rates. These mechanisms can lead to sudden structural failures without the warning signs of large deflections [6].

Pitting corrosion contributes to a reduction in the cross-sectional area of the steel and a reduction in the load-bearing capacity of the structure. Stress corrosion, on the other hand, caused by an increase in the brittleness of steel, is considered more dominant than pitting corrosion [7]. However, test results and theoretical considerations prefaced in some works indicate that this type of corrosion does not occur in the case of corrosion initiated by chloride ions [8,9]. It has also been shown that because of pitting corrosion, the proportion of brittle cracks in prestressing bands increases, but the ultimate tensile strength and deformation of steel bands significantly decrease because of the combined stress concentration and reduction in the cross-sectional area of steel bands [10]. On the other hand, based on experience as well as field tests, it can be concluded that most of the damage and failure of prestressed structures is caused by pitting corrosion of steel. However, not enough research has been performed to study the characteristics of pitting corrosion in prestressed structures, depending on the progress of corrosion processes [11,12]. It is likely that the two causes not infrequently occur together in damaged prestressed structures, so regardless of the cause of the rupture of steel strands, more studies of prestressed structures are needed to better understand these mechanisms and, as a result, prevent sudden structural failure.

To date, more studies have been conducted on the corrosion of ordinary reinforcing steel in concrete than on prestressing steel. Although the basic science of corrosion is the same for both steels, additional experimental studies are needed because of the mechanical and physical differences between them, in particular the high stress level in the prestressing steel strands and the microstructure of the prestressing steel strands. Fewer corrosion craters were observed on the surface of the compression strands than on the surface of the plain deformed rods. This smaller number of corrosion craters causes a faster increase in the local corrosion rate in these craters. In each wire and strand, pitting corrosion dominates because less water and oxygen are needed to sustain the corrosion processes [13].

Analyzing the final stresses of the tested samples, it can be seen that due to fatigue, the bond strength of the sample improves, but this improvement is not infinite. After exceeding a certain fatigue threshold, the final bond strength of the sample will deteriorate [14].

Corrosion of strands in prestressed concrete contributes more to the degradation of the tensile strength of these strands than to their bond strength to concrete. Based on experimental studies, a linear relationship between the strength of prestressed strands and the corrosion rate was determined [15].

An important issue evident in the research [16,17,18,19,20] was a marked reduction in concrete strain at the failure of corroded beams compared to reference beams, which may indicate a marked loss of ductility in corroded prestressing tendons, commonly associated with hydrogen embrittlement and stress corrosion scratching in high-strength steel.

The rate of corrosion processes is also affected by the magnitude of the applied compressive force. In general, it can be said that in an environment containing chloride ions, the larger the value of the compressive force, the higher the corrosion rate of the compressing steel [13].

Under conditions of a fixed temperature and relative humidity, the corrosion rate of rebar decreases over time in a non-linear manner. In the corrosion process, corrosion products (for example, red rust) may have gradually filled the pores near the steel surface, which reduced the amount of water in the pores and led to a decrease in the corrosion rate [13].

The corrosion rate of reinforcing steel also depends on the moisture content of the concrete. The degree of moisture impacts both the resistivity of the concrete and the rate of oxygen diffusion. Resistivity is high in dry concrete, which means that the corrosion rate is slow. Resistivity in water-saturated concrete is low, but also the oxygen diffusion rate is slow compared to the diffusion rate in dry concrete [21].

The effect of temperature on the corrosion of steel in concrete can be quite complex because temperature influences the change in the properties of the pore fluid solution and the kinetic parameters of corrosion (e.g., equilibrium potentials, exchange current densities and Tafel gradients) [22].

On one hand, a study conducted by Jaśniok and Jaśniok [23] showed a very noticeable effect of temperature change on the corrosion rate of steel, and the rate of change was closely related to the rate of temperature change. On the other hand, the effect of temperature on the corrosion rate of steel can be considered in terms of moisture availability: high temperatures clearly increase the risk of corrosion in moist concrete with chlorides, while they have the opposite effect on dry or semi-dry concrete [24].

A lot of research work has been carried out [24,25,26,27]; however, the results of these studies are difficult to analyze due to different test conditions, concrete composition and shape of samples, as well as different types of temperature and humidity distribution. Various electrochemical testing methods were also used, which can have a great impact on the test results. All these tests have been conducted in concrete reinforced with ordinary steel. However, there is a lack of research and analysis conducted on prestressed elements.

Studies are also being carried out on the influence of corrosion on the forces in the anchorage of the prestressing strings, but the specimen sizes are evidently smaller than those in actual engineering structures, and the number of specimens is small. In addition, the secondary anchorage bond performance of residual stress after the end corrosion fracture of steel strands in practical engineering remains to be further investigated [28].

Whilst considerable research has been undertaken on steel corrosion in concrete, it is more focused on reinforcing steel than on prestressing steel. Although the underlying corrosion science is the same for both steels, the applicability of data on corrosion obtained from reinforcing steel to prestressing steel needs proof due to primarily the mechanical and physical differences between the two, in particular, the high level of stresses in prestressing steel strands and the microstructure of prestressing steel strands. There are fewer corrosion cells on the surface of prestressing strands than on that of deformed bars. This reduced number of corrosion cells leads to faster growth of local corrosion at already corroded spots (cells). Since there is relatively more supply and less demand of water and oxygen for a small number of corrosion cells to sustain, pitting corrosion dominates in each strand (wire). Therefore, for corroded strands, more local and uneven corrosions occur.

From the analysis of the ultimate bond stress of all the specimens before, the presence of fatigue does improve the ultimate bond strength of the specimen, but this improvement is not infinite. When the fatigue accumulates to a certain extent, the ultimate adhesion of the specimen will be degraded.

Corrosion of the strand in a pretensioned prestressed concrete structure was found to degrade the tensile strength of the strand instead of its bond strength. The experimental investigation into the tensile behavior of corroded strands presents an approximately linear degradation law for the tensile strength of strands with the corrosion rate.

In the present study, in order to analyze the effect of temperature increase on the development of corrosion processes in prestressed concrete beams, two series of tests on the corrosion rate of reinforcement were carried out using the LPR linear polarization resistance electrochemical method. At first, tests were conducted without the influence of factors causing corrosion of steel in concrete and later with the acceleration of the process of migration of chloride ions by means of a constant electric field. Next, tests were carried out on beams stored in two types of environments differing in humidity and temperature. Four beams, two of which had previously been subjected to chloride ion migration, were placed in a climate chamber in which conditions of higher temperature (T = 30 °C) and humidity (H = 90%) were provided. Two of these four beams (one after migration and one in the initial condition) were additionally subjected to loading to maintain the scratching induced earlier. Another four analogous beams were placed under laboratory conditions in an environment of 20 °C and 30% humidity.

### Significance and Novelty of Researche

In the natural environment, there are usually both varying humidity and temperature conditions, with the simultaneous effect of load on prestressed structures often causing scratching of the structure. Therefore, it is very important to comprehensively study such situations, considering the synergistic effects of environmental factors. Especially in an environment additionally exposed to the aggressive action of chloride ions. Most corrosion studies have been conducted on small test pieces or analyzed the impact of environmental factors independently. Therefore, there is a lack of research information needed to accurately analyze the progress of corrosion processes in elements of larger size and under the influence of different environmental conditions.

## 2. Materials and Methods

The tests were performed on 10 prestressed beams of 72 × 120 × 1200 mm, reinforced with one prestressing tendon Ø6.85 mm, made of C40/50 concrete. The strands were made of Y2060-S7 steel with a characteristic tensile strength of 2060 MPa—Figure 1.

The specimens were prepared at the Konbet prestressed elements factory in Poznań, Poland. Beams of this type are usually used as lintels, while these were made with a single reinforcing strut on special order for the present study. Tests on the strength properties of concrete and steel were performed at the plant. The rest of the tests were conducted at the Laboratory of Civil Engineering of the Silesian University of Technology. Concrete with a w/c = 0.3 (the ratio of the effective water content to the cement content in a concrete mix), based on Portland cement CEM I 42.5 R (260 kg/m^3^) and natural rounded aggregate at 0–2 mm (800 kg/m^3^), as well as gravel at 2–8 mm (800 kg/m^3^), was tested. The beams immediately after fabrication at the factory Konbet (Poznań, Polska) were delivered to the Building of Laboratory of the Silesian University of Technology in Gliwice and stored in the conditions of the laboratory (humidity of about 30% and temperature of about 20 °C).

Properties and compressive strength of the used concrete mixtures are presented in Table 1.

The detailed chemical composition and basic properties (according to the producer’s specification) of the cement are given in Table 2, and the detailed compositions of concrete mixes are presented in Table 3.

The detailed chemical composition (according to the producer’s specification) of the prestressing steel is given in Table 4.

In order to analyze the effect of temperature increase on the development of corrosion processes in prestressed concrete beams, polarization tests were carried out using linear polarization resistance (LPR) methods. The tests were performed in a three-electrode system, using a Gamry Reference 600 control-recording potentiostat device (Gamry Instruments, Warminster, PA, USA) together with Framework computer software for recording LPR tests and software for analyzing the results (Gamry Echem Analyst Software, Version V).

Polarization tests of the corrosion rate of reinforcement in prestressed concrete beams were carried out in three measurement cycles at an interval of thirty days. The interval of thirty days between measurements was determined based on migration tests of concrete samples taken from the tested elements and the determination of the time after which the concentration of chloride ions at the steel surface reaches the value at which corrosion can occur [29]. Subsequent measurements were repeated at the same intervals. Immediately before the test, the beams were soaked in tap water to improve the concrete’s conductivity. The test program included three series of corrosion rate measurements on eight elements, four elements each for two measurement environments. To illustrate, beam B1 was subjected to migration of chloride ions, then loaded until the scratch was obtained, and then placed in a climatic chamber at an exposure to a temperature of 30 °C and humidity of 90%. The scratch condition was preserved at the same width with a specifically designed steel frame. After three measurement cycles, the beam has not yet been destroyed, since there are plans to continue the process of studying the development of the corrosion process. Beam B2 was also placed in a climate chamber at an exposure to a temperature of 30 °C and a humidity of 90% after the migration process but has not been subjected to loading. The beam has also not yet been destroyed. Beam B3, after the migration process, was placed in laboratory conditions where it has been exposed to a temperature of 20 °C and humidity of 30%; the beam, after three measurement cycles, has also not been destroyed. The B5 beam was not subjected to chloride migration, but it was loaded until the scratch was obtained and placed in a climatic chamber at an exposure to a temperature of 30 °C and a humidity of 90%; the beam was not destroyed after three measurement cycles. Beams B8 and B9 were destroyed in a testing machine to determine the strength of concrete prestressed beams and the dependence of deformation on the value of the applied load. The individual impacts to which each of the beams was subjected are presented in Table 5, in which the sign (+) indicates the type of impact applied to the beam, while the sign (−) indicates that the type of impact was not applied.

### 2.1. Accelerated Process of Chloride Migration in Concrete

#### 2.1.1. Preparation of Samples for Migration Tests

First, the prestressed beams were sanded and cleaned of any irregularities. Then, the reinforcement tendons were exposed in order to connect electrodes to them. The sides of the beams were painted with insulating paint to ensure one-way migration of chloride ions through the concrete.

In the next stage, 4 tanks for NaCl solution were constructed from shuttering plywood with dimensions of 17 × 250 × 1500 mm. The shuttering plywood ensured adequate rigidity and tightness of the tanks. Figure 2a shows the finished container 1 to be mounted on the prestressed concrete beam 2. The finished containers were attached to the wider surface of the beams on the side of the smaller concrete lagging, and then the edges of the joint were sealed using a flexible polymer adhesive. In Figure 2b, the finished samples prepared for migration testing are shown placed in a sealed container 3 made of Plexiglas plates.

#### 2.1.2. Migration Tests

Migration tests were conducted on four prestressed beams. Prior to testing, the test pieces were stored in distilled water for three days to increase the electrical conductivity of the concrete. Then, the test samples were connected to the migration system for a month.

The first set of two compressed beams (*1*) was placed on a large rectangular electrode (anode), made of titanium mesh covered with a layer of platinum (*2*) and boiled in distilled water (*3*) filling to a height of about 3 cm a large tank made of Plexiglas plates (*4*) in which the entire test system was placed. Containers made of shuttering plywood previously fixed to the upper surface of the beams (*5*) were filled with a 3% NaCl solution to a height of about 12 cm. The authors of the paper [30] concluded that research indicates that in a typical cell system with a volume of about 350 mL and a current voltage of 12 V, the concentration value of the source solution should not be less than 0.2 M, because below this concentration the sum of chlorides is not sufficient to ensure optimal efficiency of electric charge transport by chloride ions. The maximum value of the transfer number of chloride ions increased with an increase in the value of the concentration of the source solution used. This relationship was observed up to a concentration of 0.2 M of the source solution. Above the concentration of 0.2 M, the value of the chloride ion transfer number remained constant. It was decided to adopt a seawater concentration of about 3% (0.5 M) close to that of a natural sea.

A rectangular stainless steel electrode (cathode) was placed on top of each sample inside each container (*6*) with dimensions of 12 × 120 cm adapted to the opening of the tank. During the migration process, the electrodes do not come into direct contact with the string reinforcement because they are applied to the external surfaces of the tested element. The test set was powered by a DC (*7*) stabilized power supply of 18 V—Figure 3.

#### 2.1.3. Control of the Corrosion Process in the Process of Accelerated Migration

Before subjecting concrete to accelerated chloride migration through an electric field, polarization tests using the LPR method were conducted on all specimens to ascertain the corrosion potential of tension strings in their passive state. The chloride migration process was conducted for four beams over 30 days to observe the corrosion progression by measuring the corrosion potential. Electrochemical measurements were conducted 3 days after discontinuing the electrical supply to prevent polarization of the tested reinforcement [31].

The measurements followed a three-electrode setup, employing a tension string as the working electrode (*1*). Counter electrode (*2*), made of a stainless steel sheet in the shape of a circle with a hole in the center, was applied to the surface of the tested prestressed beams (*3*) on the side of the thinner concrete lagging. Reference electrode (*4*) of Cl^−^/AgCl,Ag composition was placed in the hole of the counter electrode on the surface of the test beam. Reference electrode (*4*) together with counter electrode (*2*) were applied at three measurement points: P1—10 cm from the left edge of the beam; P2—in the middle of the beam span; and P3—10 cm from the right edge of the beam. Point P1 was chosen at a distance of 100 mm from the edge of the beam at a place where during storage they are supported and the stresses of steel from dead weight are close to 0. Similarly, point P3 was chosen at the same distance from the opposite end to average the values of measurement results obtained. Point P2 was chosen in the middle of the beam span at a place where the prestressing steel is most stressed. In summary, it was decided to choose points P1 and P2 because of the importance of obtaining corrosion current measurements at the most and least stressed points of the prestressing steel. Point P3 was chosen to average the measurements. Since electrochemical measurements are dependent on humidity and temperature, all electrochemical studies were conducted under the same conditions. Firstly, the specimens were immersed in water for 72 h in order to stabilize the half-cell potential and avoid overload in the potentiostat. In this case, the corrosion rate was not controlled by oxygen diffusion to the steel surface. After being taken out of the water, the specimens were connected to the potentiostat (*6*), and changes in gradually stabilizing potential were observed with the reference electrode (*5*) for 60–120 min. When potential changes were at the level of 0.1 mV/s, LPR methods were performed on the steel reinforcement in concrete. The reinforcement was polarized at a rate of 1 mV/s within the range of potential changes from −150 mV to +50 mV regarding the corrosion potential. LPR tests were performed each time using a Gamry Reference 600 potentiostat (*5*) by Gamry Instruments, Warminster, PA, USA, in the potentiostatic mode within a range of frequencies of 10 mHz–100 kHz at an amplitude of 10 mV over the corrosion potential of the reinforcement (Figure 4a–c) [31,32].

### 2.2. Determination of Scratching Condition in Prestressed Concrete Beams Using Aramis Optical System

Prior to testing, the side and bottom surfaces of the beams were painted white (Figure 5a) and then sprayed with graphite paint to cover about 50% of the surface (Figure 5b).

A 3D optical scanner with software included in the Aramis system (GOM, Braunschweig, Germany) was used to record deflections and deformations during strength tests on prestressed concrete beams. The purpose of the test was to determine the state of scratching and scratch opening in prestressed concrete beams, which uses the digital image correlation technique DIC (Digital Image Correlation, Zeis, München, Germany). The system was composed of two 6 Mpx cameras (GOM, Braunschweig, Germany) equipped with 24 mm lenses to measure the test area of 120 mm × 150 mm and 90 mm in depth. The use of an optical scanner in testing allows the scratching process to be analyzed without having to stop the load in order to take measurements [33]. The analysis of the scratching condition was carried out for four prestressed concrete beams. The test program included two beams subjected to accelerated migration of chloride ions (beam B1, beam B4) and two beams not subjected to these influences (beam B5, beam B6).

Figure 6a shows a schematic of the test stand for prestressed concrete beams. The prestressed concrete beam (*1*) was loaded with a hydraulic piston cylinder (*2*) at the center of the beam span (*1*). A reciprocating actuator (*2*) was attached to a steel frame (*3*). For the beam scratching test, the load was transferred to the test piece (*1*) by the hydraulic actuator (*2*) via the core of the frame (4) of the stand for maintenance of the structure’s scratch condition (Figure 6b).

When the hydraulic cylinder (*2*), which applied the load to beam (*1*), was released, the draw deflection of the beam was maintained via turnbuckles (*5*) in the holding structure. The loading system consisted of a steel beam (*4*), to which, via turnbuckles (*5*), were connected two-armed rod tendons (*6*). The spacing of the tendons was 500 mm. The bar tendon rested on a crossbar made of flat bar (*7*). A mechanical force gauge (*8*) was placed between the steel beam (*3*) of the device and the test piece (*1*). Loads were transferred to the test piece through roller bearings (*9*) isolated from the test piece (*1*) by Teflon washers (*10*).

The scratch condition induced in the test was preserved using a special design. The stand was designed to preserve the scratch condition after the endurance tests were completed for further testing under two environmental conditions.

The acquired measurement data from the Aramis system sensors were analyzed and evaluated for deformation and displacement in the GOM Correlate program.

### 2.3. Destructive Testing of Prestressed Beams Using the Aramis Optical System

Destructive tests of two beams free of chloride ion interaction were also carried out (beam B8, beam B9).

During destructive testing, the load P from the hydraulic cylinder (*2*) was transferred directly to the beam (*1*) (Figure 7b). During the tests, displacements and load force values were recorded using sensors (*3*) compatible with the computer (*4*). The Aramis system recorded images of the displacement field at a rate of four images per second. During the test, the Aramis (*5*) system’s cameras also recorded a monitor image that displayed measurement values from the displacement sensor and the force meter.

### 2.4. Testing the Effect of Temperature Increase on the Development of Corrosion Processes

#### Control of the Corrosion Process Under Different Environmental Conditions

Four beams were stored under laboratory conditions in a room with a temperature of 20 °C and 30% humidity. These were, in the following order: beam B3—treated with chlorides in the migration process; beam B4—treated with chlorides in the migration process and with an induced scratching condition; beam B6—without exposure to chloride ions with the load maintaining the scratch condition; and beam B10—not subjected to any impacts (control element) (Figure 8a).

The remaining four beams were stored in a climate chamber at 30 °C and 90% humidity. These were, in turn: beam B1—treated with chlorides in the migration process and with an induced scratch condition; beam B2—exposed to chloride in the process of migration; beam B5—without chloride ion interactions with the load maintaining the scratch condition; and beam B7—not exposed to any interactions (Figure 8b).

The measurements followed a three-electrode setup, employing a tension string as the working electrode (*1*). Counter electrode (*2*), made of a circle-shaped stainless steel sheet with a hole in the center, was applied to the surface of the tested prestressed beams (*3*) on the thinner side of the concrete lagging through a damp felt spacer (*4*) to ensure electrical connection. A reference electrode (*5*) of Cl^−^/AgCl,Ag composition was placed in the hole of the auxiliary electrode on the surface of the test beam. The elongated shape of the reference electrode (*5*) allows it to pass through the hole of the load elements (*6*), which presses the counter-electrode (*2*) against the spacer (*4*) and the surface of the beam (*3*) to ensure a good electrical connection. Reference electrode (*5*), together with counter-electrode (*2*) and load element (*6*), was applied at three measurement points: P1—10 cm from the left edge of the beam; P2—in the middle of the beam span; and P3—10 cm of the right edge of the beam. Each time, LPR tests were performed using a Gamry Reference 600 (*7*) potentiostat from Gamry Instruments, Warminster, PA, USA (Figure 9a–c).

The corrosion current $(i_{corr}$ (μA)) can be calculated using the polarization resistance ($R_{p}$ (kΩ)) obtained through LPR measurement, as per the Stern–Geary equations [34],(1)$$R_{p}={\left.\frac{dE}{di}\right|}_{i\to 0,\;E\to E_{corr}},i_{corr}=\frac{b_{a}b_{c}}{{2.303R}_{p}(b_{a}+b_{c})},$$
where $b_{a}$ and $b_{c}$ are constants of anodic and cathodic reactions, respectively, coefficients of rectilinear slope for segments of polarization curves—anodic $b_{a}$ and cathodic $\;b_{c}$.

The corrosion rate $(V_{r}$ (mm/year)) is derived by calculating the average cross-section loss around the circumference of the bar, measured in mm, for each operational year of the structure [35].(2)$$V_{r}=0.01159\cdot I_{corr(real)}.$$

## 3. Results and Discussion

### 3.1. Results of the Accelerated Corrosion Process

LPR research was carried out on four research elements: B1, B2, B3, and B4. The first measurement was the M0 reference measurement taken before the migration of chlorides into the concrete. Measurements in the test beams were taken at three points, P1, P2, and P3, and are presented in Appendix A (Table A1).

During the entire research period, a total of 24 polarization curves were obtained. The shapes of selected ones measured at point P1 of beams B1, B2, B3, and B4 are shown in Figure 10.

Figure 11 presents a comparison of results from 24 measurements of corrosion current density (*i_corr_*_)_ of the reinforcement strings in beams B1, B2, B3, and B4 measured at three points (P1, P2, and P3) of these beams before the migration process (M0) and after 30 days of chloride ion migration (M1). The following criteria can be adopted to determine the degree of corrosion rate based on the measurement of corrosion current density: passive state for (*i_corr_* < 0.5 µA), irrelevant corrosion for (0.5 < *i_corr_* < 2 µA), low corrosion for (2 < *i_corr_* < 5 µA), moderate corrosion for (5 < *i_corr_* < 15 µA), and high corrosion for (*i_corr_* > 15 µA) [34,36,37]. The first reference study (M0) indicates that the average value of the corrosion current measured at three points, P1, P2, and P3, of the beams B1, B2, B3, and B4 is *i_corr_* = 0.4 < 0.5 µA, which indicates the passive state of the tested elements. Subsequent tests (M1) performed after 30 days of chloride ion migration indicate that the average value of corrosion current intensity measured at three points, P1, P2, and P3, of the beams B1, B2, B3, and B4 is *i_corr_* = 3.8 < 5 µA, which indicates a low state of corrosion of the studied elements.

### 3.2. Results of Scratching Condition in Prestressed Concrete Beams Using Aramis Optical System

The analysis of scratching conditions was carried out for four prestressed concrete beams: B1, B4, B5, and B6. When the deflection of the beam was taken over by the holding structure after the hydraulic cylinder was relieved, the displacement at the center of the beam span B1 was 2.5 mm. The largest deformation of the test surface at this time was 1.5%. At a load of 5.1 kN, a scratch appeared in the center of the beam span at a deflection of 2.2 mm. At this point, the largest deformation of the test surface was 0.8%.

The results of measurements of displacements and deformations of the surface of the tested element—beam B1, generated from the GOM Correlate program compatible with the Aramis system, are shown in Figure 12.

Figure 13 shows the results of measurements from the displacement sensor and force meter recorded on the 3D scanner’s cameras and read in GOM Correlate software 1.31186, 2019.

When the deflection of the beam was taken over by the holding structure after the hydraulic cylinder was relieved, the displacement at the center of the B4 beam span was 2.8 mm. The largest deformation of the test surface at this time was 1.4%. With a load equal to 6.9 kN, a scratch appeared in the center of the beam span with a deflection of 2.1 mm.

When the deflection of the B5 beam was taken over by the holding structure after the hydraulic cylinder was relieved, the displacement at the center of the beam span was 2.9 mm. The largest deformation of the test surface at this time was 1.4%. With a load equal to 3.3 kN, a scratch appeared in the center of the beam span with a deflection of 2.8 mm.

When the deflection of beam B6 was taken over by the holding structure after the hydraulic cylinder was relieved, the displacement at the center of the beam span was 3.0 mm. The largest deformation of the test surface at this time was 2.1%. With a load equal to 4.7 kN, a scratch appeared in the center of the beam span with a deflection of 2.7 mm.

The results of measurements of displacements and deformations of the surfaces of the test elements B1, B4, B5, and B6 generated from the GOM Correlate program compatible with the Aramis system are shown in Table A2 of Appendix A.

Figure 14 graphically shows the values of deflections, the drawing force, and the largest strain when the beams are scratched.

Based on the strength tests, it can be concluded that in all elements the first scratches appeared in the middle of the beams’ spans.

In B1 and B4 beams, which were subjected to accelerated migration of chloride ions, the average deflection value at scratching was 30% lower than the average deflection value of the other beams without chloride addition. Also in these beams, a 46% lower value of average deflection on the surface of the elements at the time of scratching was registered. These phenomena can be explained by the loss of prestressing force caused by corrosion products induced by the action of chloride ions. These products reduce the adhesion of steel to concrete, leading to a decrease in deflection values. The content of chloride ions in concrete can also affect its mechanical properties, such as strength and modulus of elasticity, resulting in lower values of deformation of elements subjected to chloride migration [38]. As shown in reference [39]’s paper, studies of the elastic modulus of ordinary concrete indicate a slight reduction in Young’s modulus associated with the addition of chlorides directly to the concrete mixture. In contrast, a definite increase of about 42% in the value of the modulus of elasticity was obtained in lightweight concrete in samples treated with chloride ions in an electric field.

### 3.3. Results of Destructive Testing of Prestressed Beams Using the Aramis Optical System

The destruction of beam B8 was observed at the center of the span with a beam deflection of 15 mm. At the time of destruction, the maximum destructive force reached 8.8 kN. The largest deformation of the test surface was 6.3% at the time of beam destruction. At a load equal to 6.4 kN, a scratch appeared in the center of the beam span at a deflection of 3.3 mm.

The destruction of beam B9 was observed at the center of the span with a beam deflection of 15.8 mm. At the time of destruction, the maximum destructive force reached 11.9 kN. The largest deformation of the test surface was 3.9% at the time of beam failure. At a load equal to 6.3 kN, a scratch appeared in the center of the beam span at a deflection of 3.6 mm.

Figure 15 graphically shows the values of deflections, drawing force, and highest strain at the time of loss of load capacity of the beams during destructive testing of B9 beams generated from the Aramis-compatible GOM Correlante program [33].

Figure 16 graphically shows the values of deflections, the drawing force, and the highest strain at the loss of load-bearing capacity of the beams during destructive testing.

Figure 17 shows graphs of the dependence of the displacement of the point with the greatest deformation on the value of the load increment read from the displacement sensor and the force meter recorded in beams B8 and B9 in destructive tests.

In the destructive tests, the difference between the failure forces in the tested beams was 3 kN. In order to accurately assess the load-carrying capacity of prestressed concrete beams, tests would have to be carried out on a larger number of elements.

### 3.4. Results of Corrosion Measurement in the Climate Chamber

LPR research was carried out on eight research elements: B1, B2, B3, B4, B5, B6, B7, and B10. The first measurement in beams B5, B6, B7, and B10 was the reference measurement M0. The next measurement was a control measurement, M1, taken after 30 days of storing the beams B5 and B7 in a climate chamber under conditions of 30 °C and 90% humidity. The next measurement was an M2 control measurement taken after 60 days of storing the beams B5 and B7 in a climate chamber. At the same time, there was a measurement taken on beams B6 and B10 placed in the laboratory at temperature T = 20 °C and humidity H = 30%. In beams B1 and B2, previously subjected to chloride ion migration, M2 was measured after 30 days of storing these elements in the climatic chamber and another M3 after 60 days of storing these elements in the climatic chamber. At the same time, measurements were also taken of beams B3 and B4 placed in the laboratory at T = 20 °C and humidity H = 30%. Each time, measurements in the tested beams were made at three points, P1, P2, and P3, and are presented in Appendix A (Table A3 and Table A4).

During the entire research period, a total of 84 polarization curves were obtained. The shapes of selected measures in point P1 beams B1, B2, B3, B4, B5, B6, B7, and B10 are shown in Figure 18. For the set of results the graphs are based on, refer to the dataset [40].

Figure 19 shows a comparison of the results of 36 corrosion current density measurements, i_corr_, of the reinforcement strings in beams B1, B2, B3, and B4 measured at three points, P1, P2, and P3, of these beams after 30 days of chloride ion migration (M1), after 30 days of storage of B1 and B2 beams in a climate chamber at 30 °C and 90% humidity (M2), and after 60 days of storage of beams B1 and B2 in a climate chamber at 30 °C and 90% humidity (M3). Measurements of beam B1 stored in the chamber were juxtaposed with measurements of beam B4 stored in the laboratory at temperature T = 20 °C and humidity H = 30%, taken at the same time, as both beams were loaded in order to preserve the scratching of these beams. On the other hand, the measurements of beam B2 stored in the chamber were contrasted with those of beam B3 stored in the laboratory at T = 20 °C and H = 30% humidity.

The first test (M1) shows that the average value of the corrosion current measured at the three points, P1, P2, and P3, of beams B1, B2, B3, and B4 is *i_corr_* = 2 < 3.8 < 5 µA, which indicates a low state of corrosion of the test items. Subsequent tests (M2) of unscratched beam B3 stored in the laboratory at temperature T = 20 °C and humidity H = 30% indicate that the mean value of the corrosion current decreases to the value *i_corr_* = 1.4 µA, which indicates an irrelevant level of corrosion hazard. On the other hand, another averaged measurement (M3) taken after another 30 days shows a fivefold increase in corrosion and is *i_corr_* = 6.9 < 15 µA, which already indicates an average level of corrosion. When considering the distribution of corrosion current in this beam, the highest value was obtained in the middle of the beam span at point P2 and was *i_corr_* = 9.7 µA. In contrast, subsequent tests (M2 and M3) of the unscratched B2 beam stored in a climate chamber at 30 °C and 90% humidity indicate that the mean value of the corrosion current remains very low, *i_corr_* = 0.7 µA, close to passivation of steel. Storing the beam in high humidity conditions extinguished the corrosion processes previously initiated by the presence of chloride ions. Therefore, it can be assumed that the decisive factor affecting the increase in the rate of these processes is the access of oxygen, hindered in this case by the saturation of concrete with moisture. When the cover thickness increases to 20 to 40 mm, the lack of actual oxygen flux relative to the necessary oxygen content indicates that the corrosion process is oxygen limited [41]. The higher values of corrosion current observed in uncracked beams compared to cracked beams and beams with additional loading occurring both under laboratory conditions and in the climatic chamber can be attributed here to changes in the corrosion rate caused by the effect of stresses acting on the beam. Since the reinforcing string is in tension in the prestressed concrete member, we can expect corrosion processes to be faster in it than in ordinary reinforcement. On the other hand, once scratching is achieved in a prestressed concrete element, the prestressing forces are reduced, which contributes to a decrease in the speed of corrosion processes.

On the other hand, in scratched beam B4 stored in the laboratory at temperature T = 20 °C and humidity H = 30%, a subsequent test (M2) showed that the mean value of the current intensity was *i_corr_* = 5.7 > 5 µA, which could indicate a medium corrosion hazard. However, in a subsequent test (M3), the average value of the corrosion current was *i_corr_* = 3.0 µA, which again indicated a low level of corrosion hazard. The mean value of the corrosion current intensity (*i_corr_* = 1.6 µA) in scratched beam B1, which had been stored in the climatic chamber (30 °C and 90%), was approximately three times lower than the average value of the corrosion current (*i_corr_* = 4.4 µA) in scratched beam B4 stored under laboratory conditions at 20 °C and 30% humidity. The slower corrosion process in the climatic chamber, despite the higher temperature, may be due to the poorer oxygen supply caused by the higher humidity level in the chamber. If we compare the corrosion rate in two media with high moisture content or immersed directly in a solution, we observe an increase in the corrosion rate with increasing temperature. On the other hand, in semi-dry or low-moisture concrete, we observe a decrease in corrosion processes with an increase in temperature. If we superimpose both factors, we find that with very high moisture content or total immersion, corrosion processes occur more slowly than in semi-dry or low-moisture samples. The decisive aspect of the corrosion process here is the access of oxygen, which is easier in samples with lower humidity than in samples with high humidity. A similar effect was observed in the work [42], where corrosion potential decrease and corrosion mass loss at 50 °C in comparison to 40 °C temperature conditions were observed. It can possibly be explained by the reduction of oxygen solubility in the pore solution at high temperature, resulting in an oxygen-controlled corrosion reaction at high chloride concentration. Another reason can be the blockage of concrete pores at high relative humidity and high temperature, resulting in discontinuity of interconnected concrete, which in turn results in the shortage of oxygen at such a high humidity and high temperature condition.

Figure 20 presents a comparison of results from 36 measurements of corrosion potential obtained from *E_corr_* of the reinforcement strings in beams B1, B2, B3, and B4 measured at three points, P1, P2, and P3, of these beams after 30 days of chloride ion migration (M1), after 30 days of storage of beams B1 and B2 in a climatic chamber at 30 °C and 60% humidity (M2), and after 60 days of storage of beams B1 and B2 in a climatic chamber at 30 °C and 60% humidity (M3).

The following criteria can be adopted to determine corrosion probability based on corrosion potential measurements: 95% corrosion probability for (*E_corr_* < −443 mV), 50% corrosion probability for (−443 < *E_corr_* < −293 mV), 5% corrosion probability for (*E_corr_* > −293 mV) [34,36,37]. The first tests (M1) indicate that the average value of the corrosion potential measured at the three points, P1, P2, and P3, of beams B1, B2, and B3 is *E_corr_* = 650 > 443 mV, indicating a very high (95%) probability of corrosion of the test pieces. In contrast, the first test (M1) of beam B4 indicates that the average value of the corrosion potential measured at the three points, P1, P2, and P3, is *E_corr_* = 52 < 293 mV, indicating a very low (5%) probability of corrosion of the test pieces. Subsequent tests (M2 and M3) of all beams B1, B2, B3, and B4 indicate that the average corrosion potential value was *E_corr_* = 443 > 380 > 293 mV, which could indicate a medium (50%) corrosion risk.

Figure 21a presents a comparison of results from 36 measurements of corrosion current density, i_corr_, of the reinforcement strings in beams B5, B6, B7, and B8 measured at three points, P1, P2, and P3, of these beams before placing the beams B5 and B7 in a climate chamber (M1), after 30 days storage of beams B5 and B7 in a climate chamber at 30 °C and 60% humidity (M2), and after 60 days storage of beams B5 and B7 in a climate chamber at 30 °C and 60% humidity (M3). Measurements of beam B5, stored in the chamber, were juxtaposed with measurements of beam B6, stored in the laboratory at T = 20 °C and humidity H = 30%, taken at the same time, as both beams were loaded in order to preserve the scratching of these beams. Meanwhile, the measurements of beam B7 stored in the chamber were compared with those of beam B10 stored in the laboratory at T = 20 °C and humidity H = 30%.

Figure 21b presents a comparison of results from 36 measurements of corrosion potential obtained from the *E_corr_* of the reinforcement strings in beams B5, B6, B7, and B8 measured at three points, P1, P2, and P3, of these beams before placing beams B5 and B7 in a climatic chamber (M1), after 30 days of storage of beams B5 and B7 in a climatic chamber at a temperature of 30 °C and 60% humidity (M2), and after 60 days of storage of beams B1 and B2 in a climatic chamber at a temperature of 30 °C and 60% humidity (M3).

All tests (M1, M2, and M3) indicate that the average value of the corrosion current measured at the three points, P1, P2, and P3, of beams B5, B6, B7, and B10 is icorr = 0.2 < 0.5 µA, indicating the absence of corrosion of the test pieces.

All tests (M1, M2, and M3) indicate that the mean value of the corrosion potential measured at the three points, P1, P2, and P3, of the beams B5, B6, B7, and B10 is Ecorr = 183 < 293 mV, indicating a very low (5%) probability of corrosion of the tested components.

It is difficult, on the basis of the results of the B5, B6, B7, and B10 beams not previously subjected to the process of migration of chloride ions into the interior of the beams, to draw unambiguous conclusions for now. It is necessary to carry out corrosion tests for a longer period of time because, as is known, corrosion processes occur very slowly, especially in the absence of a direct corrosion-initiating factor in the form of chloride ions. It can be noted that in the scratched B6 beam there was an increase in the average value of the corrosion current in the third measurement, which may suggest an increase in this value at a later stage of the study.

The results obtained, although they are still the result of preliminary studies, can be of great importance in the process of numerical modeling of the influence of climatic conditions such as temperature and humidity on the development of corrosion processes in prestressed concrete elements. It should be noted in the process of simulation of corrosion processes that the increase in the rate of these processes is more influenced by the possibility of oxygen supply than by an increase in the humidity of concrete.

Continuation of the initiated research will allow, in the next stage, to study the effect of corrosion rate on the strength of prestressed concrete elements. This information will be used to develop a method for predicting the load-carrying capacity of prestressed concrete elements and taking this influence into account in the structural design process.

## 4. Conclusions

Based on the conducted tests, the following conclusions can be drawn:The use of the chloride ion migration method proved to be an effective method for initiating and accelerating corrosion processes in all test items subjected to this process.Based on the strength tests, it can be concluded that in all elements the first scratches appeared in the middle of the span of the beams.A very important and so far overlooked issue is the effect of chloride ion content on the mechanical properties of concrete. The content of chloride ions in concrete can contribute to changes in the elastic and strength properties of concrete. A 30% decrease in deflection values and a 46% decrease in strain values in beams subjected to accelerated chloride migration before the test may indicate, on the one hand, the effect of a change in the adhesion of reinforcing strings due to corrosion and, on the other hand, a change in the mechanical properties of concrete.In order to accurately assess the load-bearing capacity of prestressed concrete beams, tests would have to be conducted on a larger number of elements.An increase in temperature is not the only condition implying an increase in the corrosion rate in both scratched and additionally loaded elements and unloaded elements. More important is the ease of access to oxygen, which, when the environment is very humid, is inhibited probably by filling the concrete pores with water.The conducted tests should be extended in order to observe the progress of corrosion processes over time, perhaps with a simultaneous change in the environmental conditions set in the climate chamber.It is also necessary to perform strength tests on the beams after the corrosion tests are completed in order to further observe the influence of corrosion processes on the values of deflection and deformation on prestressed concrete beams.It is also necessary to carry out more tests to confirm the preliminary test results obtained.

The results obtained from tests carried out under complex loading conditions of both variable temperature, humidity, and long-term loading will allow more accurate prediction of the performance of the structure under natural operating conditions and predict possible risks caused by corrosion of steel. Numerical modeling of the structure’s behavior will make it possible to predict possible threats occurring under conditions of real climate and aggressive environments containing chloride ions. This will help avoid problems at the stage of designing prestressed structures.

## Figures and Tables

**Figure 1 materials-18-01553-f001:**
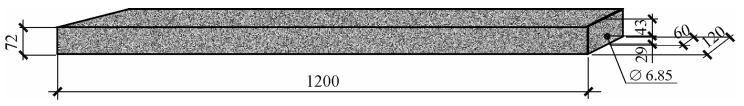
Schematic drawing of the tested prestressed beam.

**Figure 2 materials-18-01553-f002:**
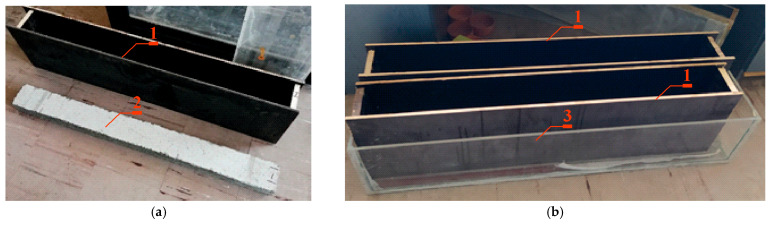
(**a**) Tank (*1*) for NaCl solution made of shuttering plywood ready to be mounted on a prestressed concrete beam (*2*). (**b**) Samples prepared for migration tests placed in a sealed tank (*3*) made of Plexiglas panels.

**Figure 3 materials-18-01553-f003:**
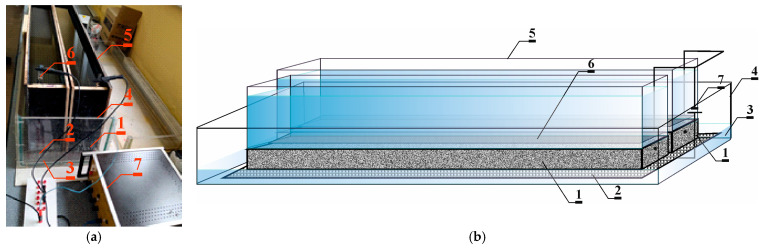
The experimental setup for accelerating the migration of chloride ions to concrete through the application of an electric field. (**a**) Research view. (**b**) Schematic image of the testing procedure: (*1*)—concrete test specimen, (*2*)—titanic anode coated with platinum, (*3*)—distilled water, (*4*)—tank made of Plexiglas plates, (*5*)—a tank made of shuttering plywood filled with 3% NaCl solution, (*6*)—stainless steel cathode, (*7*)—power source (stabilized power supply).

**Figure 4 materials-18-01553-f004:**
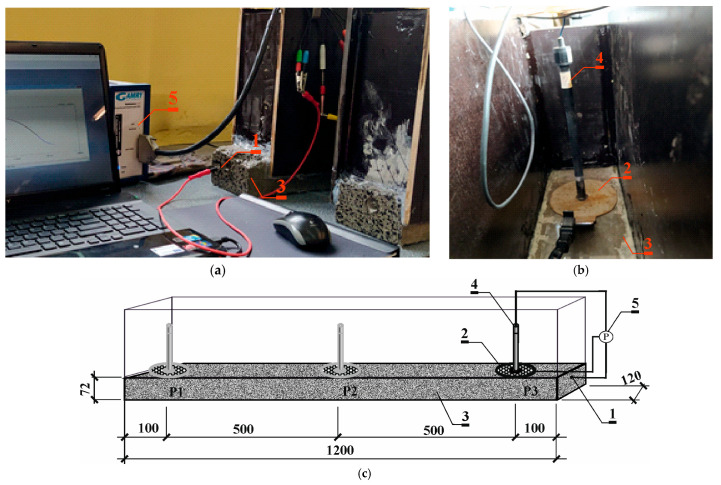
The applied test stand for polarization tests with the LPR and EIS methods: (**a**) photos of the testing procedure; (**b**) view of the chlorosilver electrode, placed in the center of the counter-electrode; (**c**) schematic image of the study: (*1*)—prestressing tendon Ø6.85 mm made of steel Y2060-S7 (working electrode), (*2*)—counter electrode, (*3*)—prestressed beam, (*4*)—(Cl^−^/AgCl,Ag) electrode as the reference electrode, (*5*)—Gamry Reference 600 potentiostat with a computer unit and Gamry software.

**Figure 5 materials-18-01553-f005:**
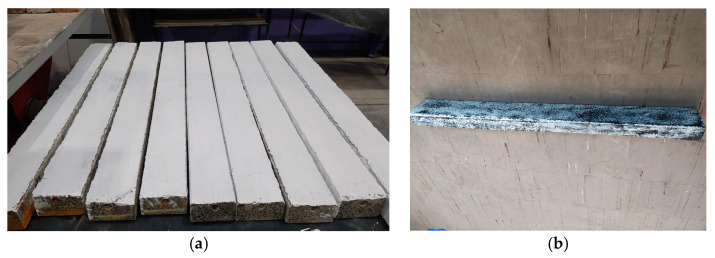
Preparation of the beams for testing (**a**) The side and bottom surfaces of the beams painted white. (**b**) beam sprayed with graphite paint.

**Figure 6 materials-18-01553-f006:**
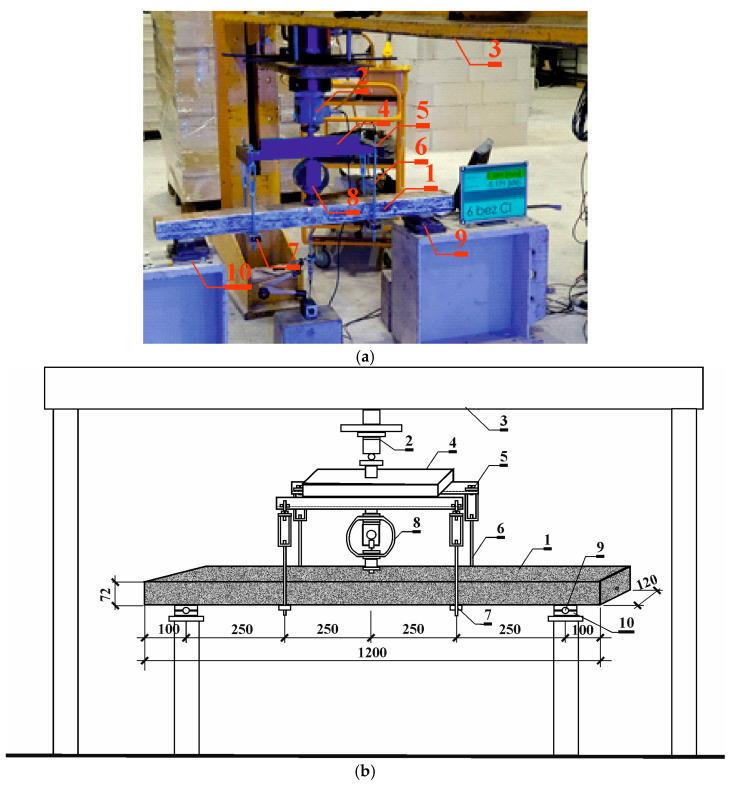
Strength test stand: (**a**) photos of the testing procedure; (**b**) schematic image of the testing procedure 1—prestressed concrete beam, 2—piston actuator, 3—steel frame, 4—steel beam, 5—turnbuckles, 6—bar ties, 7—flat bar, 8—mechanical actuator, 9—roller bearings, 10—Teflon washers.

**Figure 7 materials-18-01553-f007:**
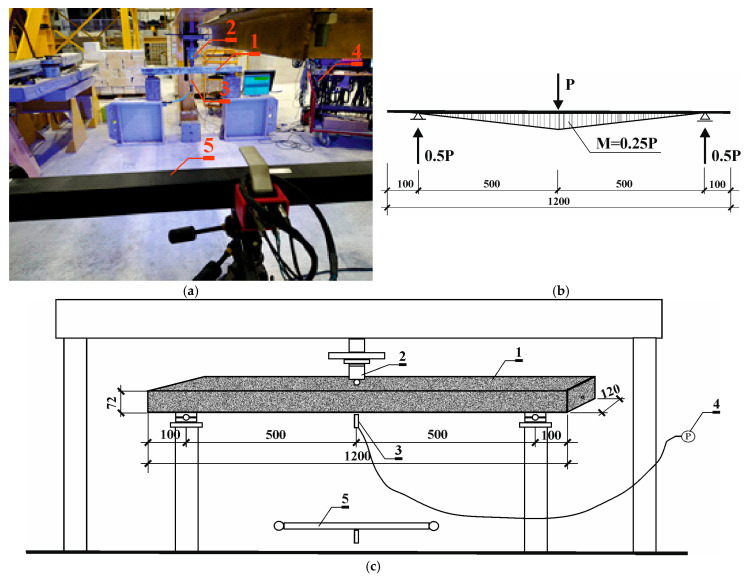
Destructive test stand: (**a**) photos of the testing procedure; (**b**) load P action diagram; (**c**) schematic of the testing procedure: (*1*)—prestressed concrete beam, (*2*)—hydraulic actuator, (*3*)—sensor, (*4*)—computer, (*5*)—Aramis system cameras.

**Figure 8 materials-18-01553-f008:**
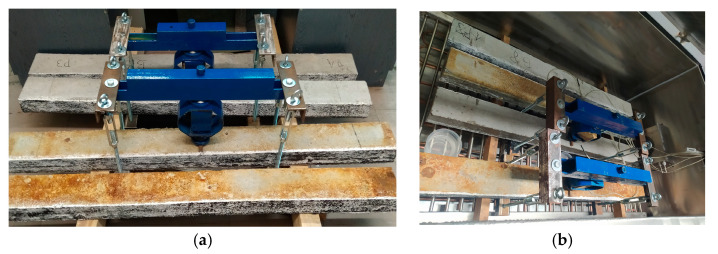
(**a**) Beams B3, B4, B6, B10 stored under laboratory conditions in a room with a temperature of 20 °C and humidity of 30%. (**b**) Beams B1, B2, B5, B7 stored in a climate chamber at a temperature of 30 °C and humidity of 90%.

**Figure 9 materials-18-01553-f009:**
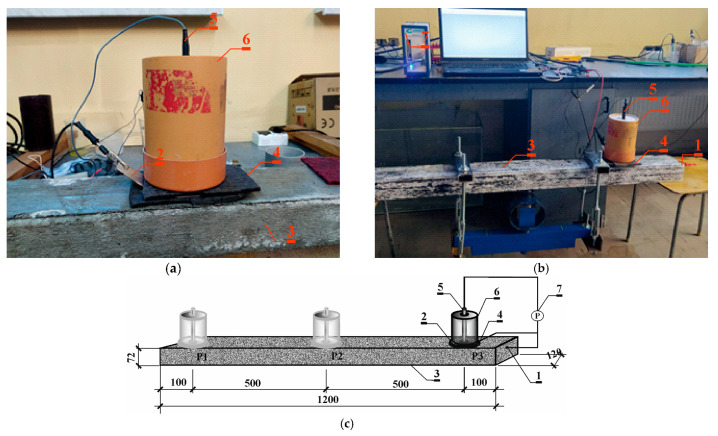
The applied test stand for polarization tests with the PR and EIS method. (**a**,**b**) View; (**c**) scheme: (*1*)—prestressing tendon Ø6.85 mm made of steel Y2060-S7 (working electrode), (*2*)—auxiliary electrode, (*3*)—compressed beam, (*4*)—moist felt spacer, (*5*)—(Cl^−^/AgCl,Ag) electrode as the reference electrode, (*6*)—load element, (*7*)—Gamry Reference 600 potentiostat with a computer unit and Gamry software.

**Figure 10 materials-18-01553-f010:**
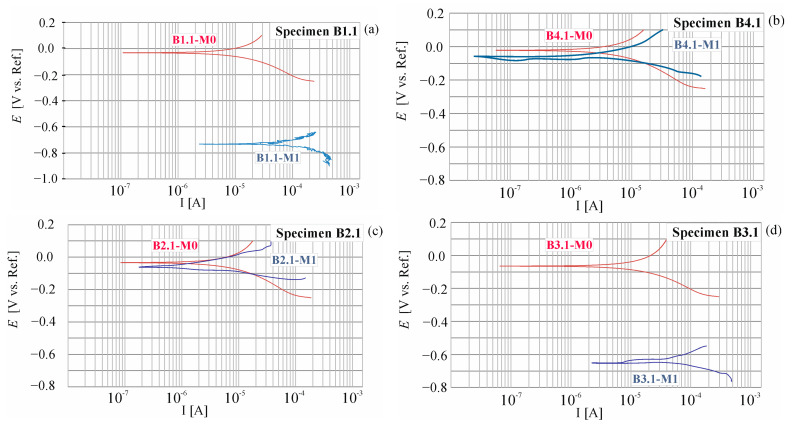
Potentiodynamic polarization curves (M0) before chloride migration and (M1) after 30 days of migration measured at the point P1 of tested beams: (**a**) beam B1, (**b**) beam B4, (**c**) beam B2, (**d**) beam B3.

**Figure 11 materials-18-01553-f011:**
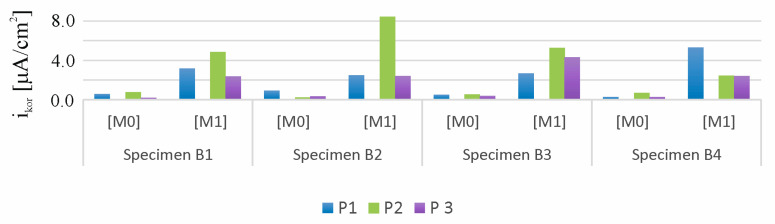
Distribution of corrosion current densities obtained for the specimens B1, B2, B3, and B4: M0—before chloride migration, M1—after 30 days of migration.

**Figure 12 materials-18-01553-f012:**
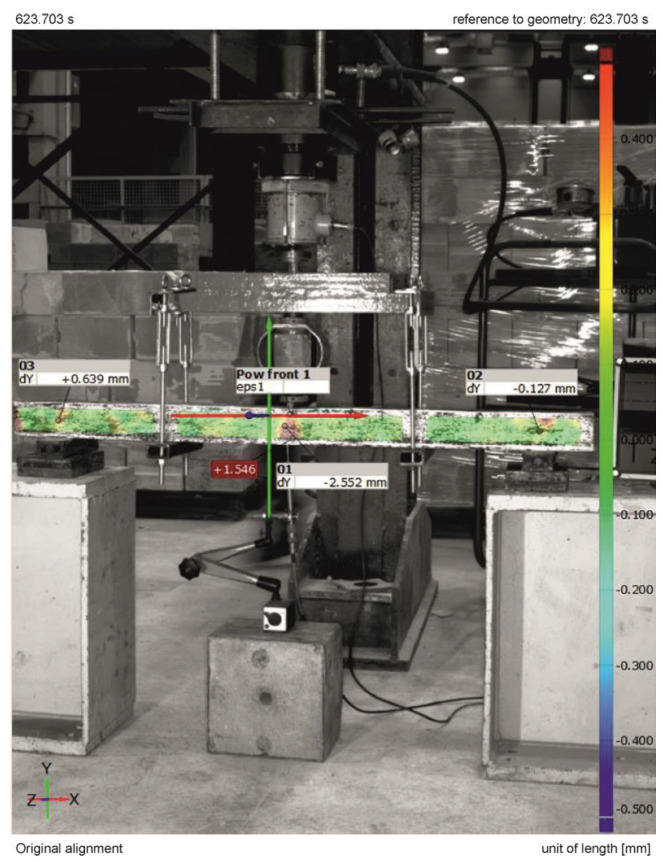
Measurement results of displacements and deformations in beam No. 1 when the deflection of the beam is taken over by the retaining structure recorded by the Aramis System.

**Figure 13 materials-18-01553-f013:**
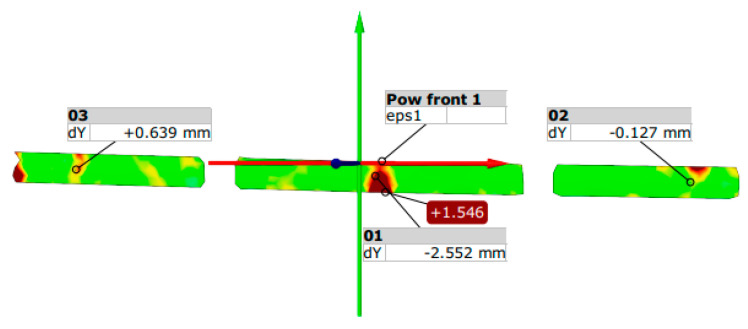
3D view of displacement and strain measurement in beam No. 1 when the deflection of the beam is taken over by the holding structure.

**Figure 14 materials-18-01553-f014:**
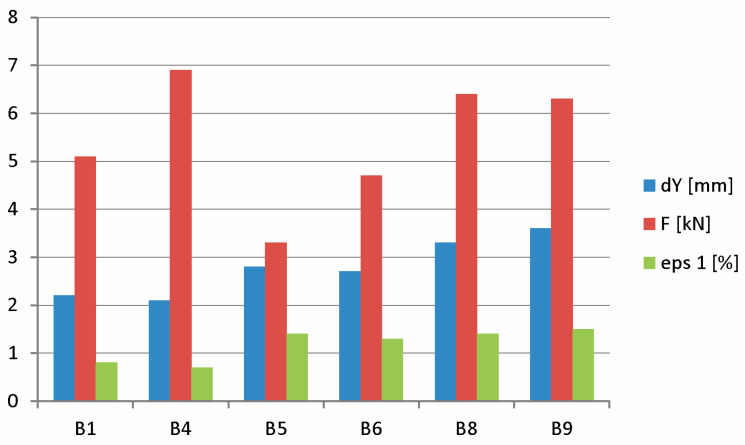
Summary of deflections, drawing force, and deformations when beams are scratched (B1, B4, B5, B6, B8, B9).

**Figure 15 materials-18-01553-f015:**
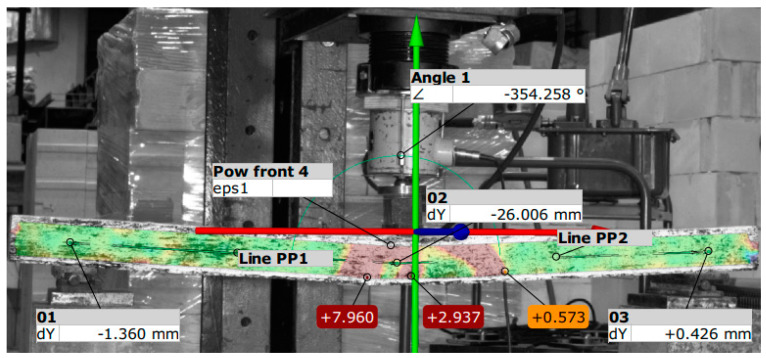
Image of the failure of the test item—beam no. 9 (system Aramis).

**Figure 16 materials-18-01553-f016:**
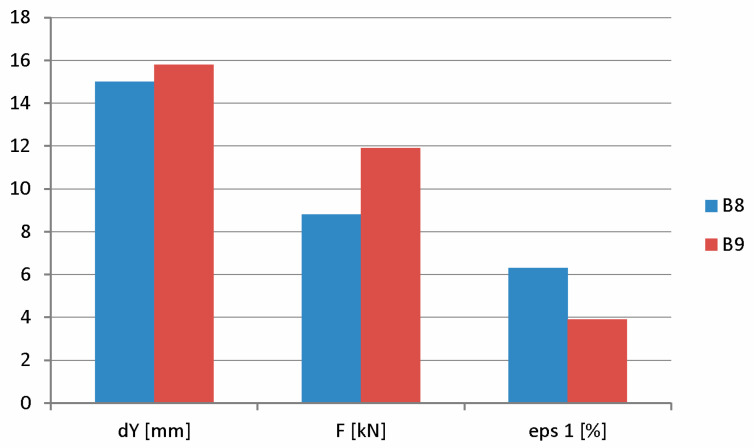
Summary of deflections, drawing force, and deformations at the moment of destruction of beams B8 and B9.

**Figure 17 materials-18-01553-f017:**
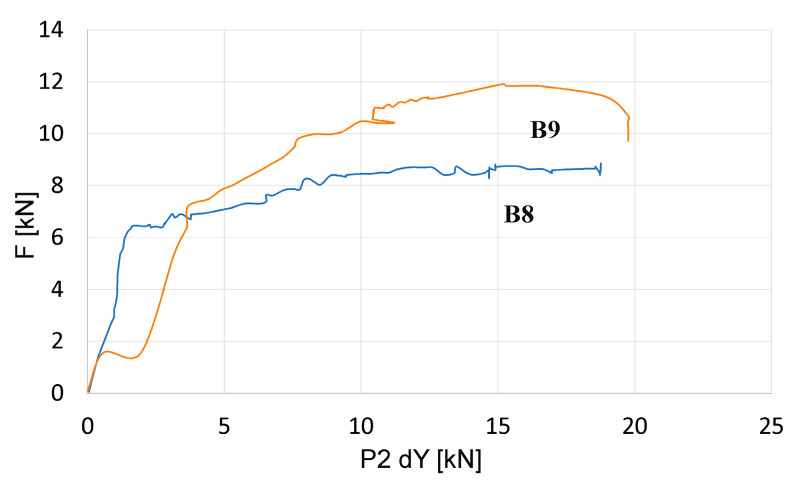
Diagram of the dependence of the displacement of the point with the largest deformation on the value of the load increment recorded during the destructive tests of beams B8 and B9.

**Figure 18 materials-18-01553-f018:**
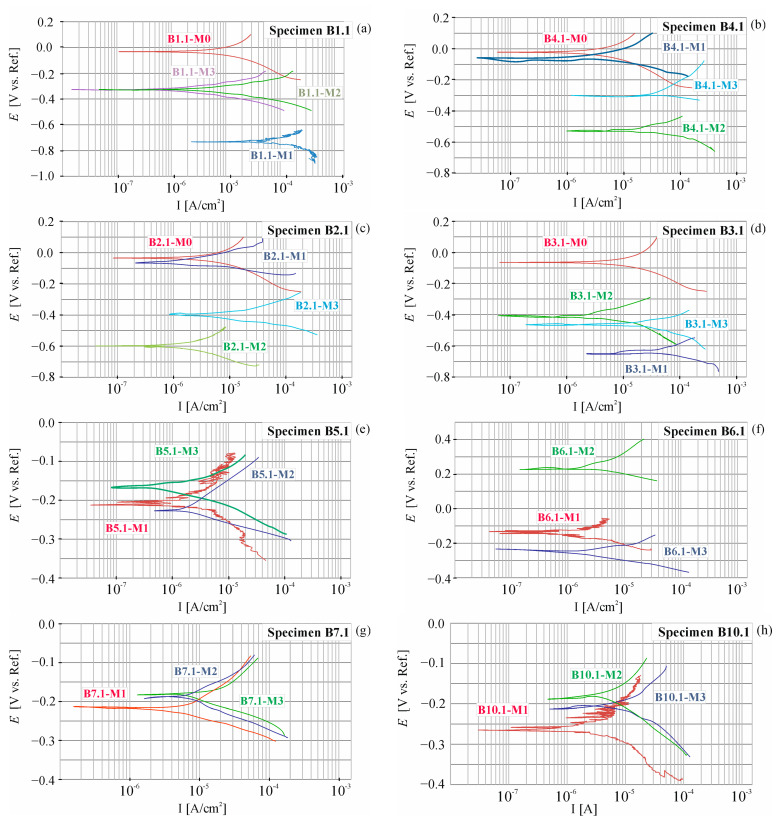
Potentiodynamic polarization curves for the beams B5, B6, B7, and B10 (M0) before placing the beams B5 and B7 in the climate chamber (cc), (M1) made after 30 days of storing B5 and B7 beams in cc, (M2) made after 60 days of storage of beams B5 and B7 in cc, and for beams B1, B2, B3, and B4 (M2) made after 30 days of storage of beams B1 and B2 at cc, (M3) made after 60 days of storage of beams B1 and B2 in cc, at point P1 of the tested beams: (**a**) B1, (**b**) B4, (**c**) B2, (**d**) B3, (**e**) B5, (**f**) B6, (**g**) B7, (**h**) B10.

**Figure 19 materials-18-01553-f019:**
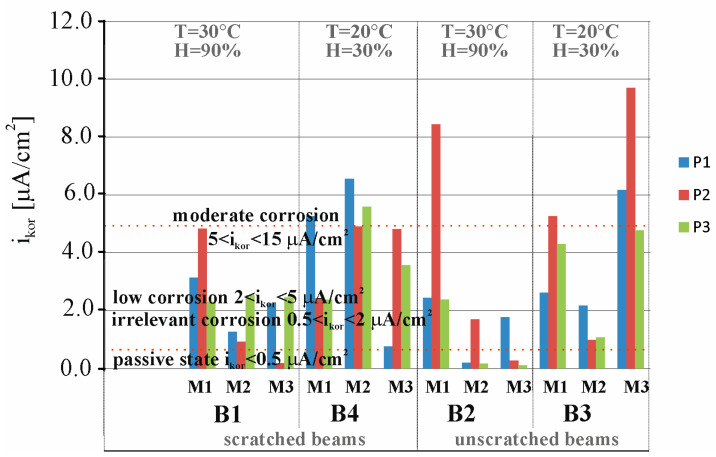
Distribution of corrosion current densities obtained for the specimens B1, B2, B3, and B4: M1—after 30 days of migration, M2—after 30 days of storing the beams B1 and B2 in the climate chamber, M3—after 60 days of storing the beams B1 and B2 in the climate chamber.

**Figure 20 materials-18-01553-f020:**
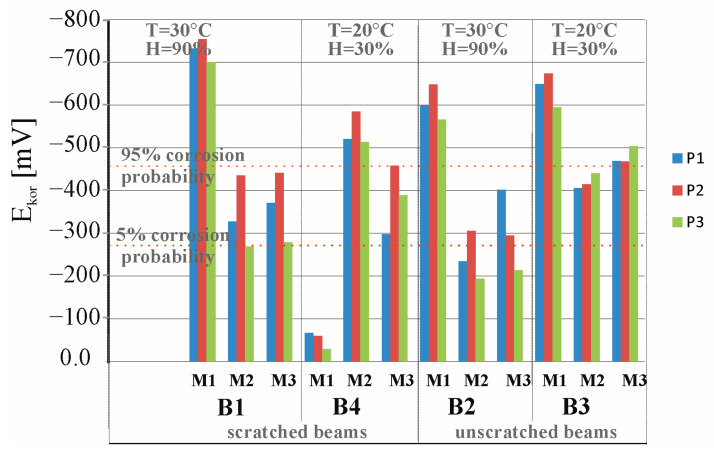
Distribution of corrosion potential obtained for the specimens B1, B2, B3, and B4: M1—after 30 days of migration, M2—after 30 days of storing the beams B1 and B2 in the climate chamber, M3—after 60 days of storing beams B1 and B2 in the climate chamber.

**Figure 21 materials-18-01553-f021:**
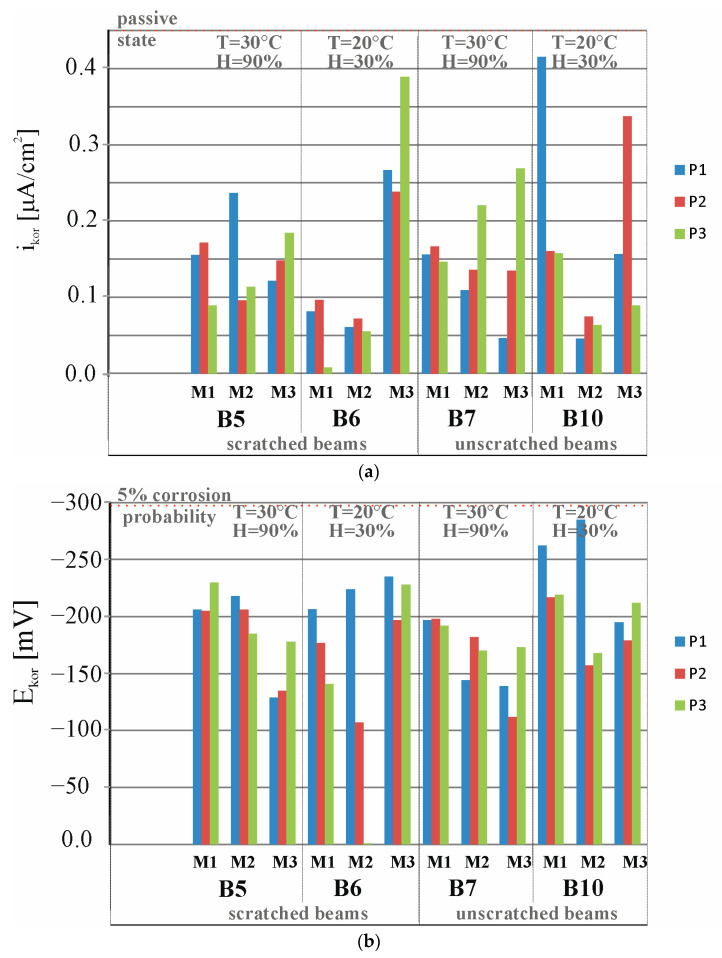
Distribution of (**a**) corrosion current densities and (**b**) corrosion potential obtained for the specimens B5, B6, B7, and B8: M1—before placing beams B5 and B7 in the climate chamber, M2—after 30 days of storing beams B5 and B7 in the climate chamber, M3—after 60 days of storing beams B5 and B7 in the climate chamber.

**Table 1 materials-18-01553-t001:** Properties and compressive strength of the concrete mixture used in the tested elements.

Compressive Strength [MPa]	Density [kg/m^3^]	Porosity [%]
58.3	2359	12

**Table 2 materials-18-01553-t002:** Chemical compositions of CEM I 42.5 R.

Constituent % mass	SiO_2_	Al_2_O_3_	Fe_2_O_3_	CaO	MgO	K_2_O	Na_2_O	Eq. Na_2_O	SO_3_	Cl
	19.38	4.57	3.59	63.78	1.38	0.58	0.21	0.59	3.26	0.069

**Table 3 materials-18-01553-t003:** Composition of studied concrete mixtures.

Sand (0–2) * mm [kg/m^3^]	Gravel (2–8) * mm [kg/m^3^]	Type of Cement	Cement [kg/m^3^]	w/c
800	800	CEM I 42.5 R **	260	0.3

* Grain diameter range; ** CEM I—Portland cement; R—high-strength early cement grade.

**Table 4 materials-18-01553-t004:** Chemical compositions of prestressing steel Y2060-S7.

Constituent % mass	C	Si	Mn	Cr	P	S
	0.7–0.9	0.15–0.30	0.60–0.90	≤0.30	≤0.035	≤0.035

**Table 5 materials-18-01553-t005:** The types of impacts to which the respective prestressed beams were subjected.

Beam Number	B1	B2	B3	B4	B5	B6	B7	B8	B9	B10
Chloride migration	+	+	+	+	−	−	−	−	−	−
Scratch condition	+	−	−	+	+	+	−	−	−	−
Placement in the climate chamber	+	+	−	−	+	−	+	−	−	−
Destruction	−	−	−	−	−	−	−	+	+	−

## Data Availability

The data presented in this study are available on Zenodo 2025 https://doi.org/10.5281/zenodo.14840580 (accessed on 9 February 2025) [40].

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
