# Peer review of "Testing the Corrosion Rate of Prestressed Concrete Beams Under Variable Temperature and Humidity Conditions"

_materials, 2025, doi:10.3390/ma18071553_

Round 1

Reviewer 1 Report

Comments and Suggestions for Authors

The topic of the present research is very important and interesting, however the manuscript must be revised to enhance its importance and quality.

The experimental methods are well described, but what I miss in the manuscript:

  1. Explanation of the experimental design (Tab. 1)
  2. Discussion of the obtained results - what is their meaning and importance.
  3. Conclusions must be revised and generalized

Minor comments:

  1. Check the affiliations (wrong numbers)
  2. First line in chapter 2: use mm, not cm
  3. Sentence "beams were soaked in tap water to improve the concreteʹs resistivity" - in my opinion, is is better to say "improve conductivity" (it sounds more logicaly) 
  4. A polish word remained in text in chapter 4.1.
  5. "All analyses and evaluation of corrosion rates were carried out based
    on the assumptions presented in the paper [22]:" - these assumptions should be briefly repeat in the paper.

Author Response

Response to Reviewer 1 Comments

1) The experimental methods are well described, but what I miss in the manuscript:

1a) Explanation of the experimental design (Tab. 1)

Ad 1a) In Table 1 I misspelled the table title, it should be: Properties and compressive strength of the concrete mixture used in the tested elements.

As for Table 3, I also supplemented the description of the experiment for individual elements in the text.

1b) Discussion of the obtained results - what is their meaning and importance.

Ad 1b) I have supplemented the discussion in the text.

1c) Conclusions must be revised and generalized

Ad 1c) I have revised and generalized the conclusions.

2) Minor comments:

2a) Check the affiliations (wrong numbers)

Ad 2a) I’ve checked the affiliations.

2b) First line in chapter 2: use mm, not cm

Ad 2b) I have corrected the mistake.

2c) Sentence "beams were soaked in tap water to improve the concreteʹs resistivity" - in my opinion, is is better to say "improve conductivity" (it sounds more logicaly)

Ad 2c) I have corrected the sentence.

2d) A polish word remained in text in chapter 4.1.

Ad 2d) I have corrected the mistake.

2e) "All analyses and evaluation of corrosion rates were carried out based on the assumptions presented in the paper [22]:" - these assumptions should be briefly repeat in the paper.

Ad 2e) I have added it in the text

Reviewer 2 Report

Comments and Suggestions for Authors

The paper „Testing the corrosion rate of prestressed beams under variable temperature and humidity conditions” is research paper. It is aimed to present results related to the effect of temperature and humidity changes on the development of corrosion processes in prestressed concrete beams”.

 This paper evidences the expertise and experience of authors in the topic of research, proven by the detailed explanation, reasonable assumption of the phenomena and their results, the experimental stands and devices (systems) used.

 Some comments on minor issues follow next.

  1. Please, if convenient, reconsider the paper title - so as to mention “(…) the corrosion rate of prestressed concrete beams”.
  2. The keywords should mention “concrete”.
  3. Please check the phrase “In addition, due to the high strength of prestressing reinforcements, they are also subject to other degradation mechanisms” (page 2). Due to should be changed by “because of”
  4. In Table 1 - explicitly mention the significance of signs “+” and “-“.
  5. At Figure 10, if the unit is SI [A], than the graphs should mention 10-6; 10-3 and so on, meaning exponents of Ampere subdivision. The same for Figure 18.
  6. Please, check and correct English, if necessary
  7. The authors are kindly asked to follow the guidelines of manuscript preparation - when refering to chapter 2 and chapter 3.

https://www.mdpi.com/journal/materials/instructions#preparation

“(…) but all manuscripts must contain the required sections: Author Information, Abstract, Keywords, Introduction, Materials & Methods, Results, Conclusions, Figures and Tables with Captions, Funding Information, Author Contributions, Conflict of Interest and other Ethics Statements”

Comments on the Quality of English Language

Please, check and correct English, if necessary

Author Response

Response to Reviewer 2 Comments

The paper „Testing the corrosion rate of prestressed beams under variable temperature and humidity conditions” is research paper. It is aimed to present results related to the effect of temperature and humidity changes on the development of corrosion processes in prestressed concrete beams”.

This paper evidences the expertise and experience of authors in the topic of research, proven by the detailed explanation, reasonable assumption of the phenomena and their results, the experimental stands and devices (systems) used.

Some comments on minor issues follow next.

1) Please, if convenient, reconsider the paper title - so as to mention “(…) the corrosion rate of prestressed concrete beams”.

Ad 1) I have corrected the paper title.

2) The keywords should mention “concrete”.

Ad 2) I have completed the keywords.

3) Please check the phrase “In addition, due to the high strength of prestressing reinforcements, they are also subject to other degradation mechanisms” (page 2). Due to should be changed by “because of”

Ad 3) I have corrected the sentence.

4) In Table 1 - explicitly mention the significance of signs “+” and “-“.

Ad 4)  I have included the explanation of significance and changed the table number from 1 to 3 because it was mistakenly given as number 1.

5) At Figure 10, if the unit is SI [A], than the graphs should mention 10-6; 10-3 and so on, meaning exponents of Ampere subdivision. The same for Figure 18.

Ad 5) I have corrected the figures.

6) Please, check and correct English, if necessary

Ad 6) I’ve checked and corrected English throughout the paper.

7) The authors are kindly asked to follow the guidelines of manuscript preparation - when refering to chapter 2 and chapter 3.

https://www.mdpi.com/journal/materials/instructions#preparation

“(…) but all manuscripts must contain the required sections: Author Information, Abstract, Keywords, Introduction, Materials & Methods, Results, Conclusions, Figures and Tables with Captions, Funding Information, Author Contributions, Conflict of Interest and other Ethics Statements”

Ad 7) I have combined the sections 2 and 3 according to the guidelines.

Reviewer 3 Report

Comments and Suggestions for Authors

The manuscript addresses the corrosion behavior of prestressed concrete beams under varying temperature and humidity conditions, incorporating chloride ion migration and induced cracking conditions. The study presents valuable experimental data; however, several aspects require major revision to enhance clarity, strengthen the methodology, and improve the overall impact and rigor of the findings. Below are detailed recommendations for revision:

  1. Clarify in the title if "pressed beams" refers specifically to prestressed concrete beams.
  2. Clearly state in the abstract the significance and novelty of your work compared to existing literature.
  3. Specify clearly how the "variable temperature and humidity conditions" differ from previous studies and their significance.
  4. Consider adding "digital image correlation" and "chloride migration" to the keywords.
  5. The description of stress corrosion cracking and hydrogen embrittlement is clear; however, briefly highlight the real-world implications or typical cases of failure.
  6. Provide more references to recent studies on prestressed structures to contextualize the novelty of your research clearly.
  7. Explicitly articulate how your study addresses the knowledge gap identified in prestressed structures compared to previous work on ordinary reinforced concrete.
  8. Justify the choice of the w/c ratio (0.3) clearly in the context of corrosion studies.
  9. Provide additional details about the storage conditions of the beams prior to testing.
  10. Clarify why a 3% NaCl solution was chosen and its relevance to realistic scenarios.
  11. Clearly mention any challenges or limitations encountered in maintaining the electrical field uniformly during the migration tests.
  12. Explain the criteria for selecting measurement intervals (30-day cycles). The basics of the methodology of LPR should be explained by citing the reference: https://doi.org/10.1016/j.corsci.2023.111118
  13. Clarify how strain measurement accuracy was verified or calibrated.
  14. Include justification for selecting only two beams for destructive tests and discuss the implications on statistical reliability.
  15. Clearly explain why specific measurement points (P1, P2, P3) were selected.
  16. Figures showing polarization curves should include clearer axis labeling and scales for enhanced readability.
  17. Clarify the reason behind classifying corrosion current density below 5 µA as "low" corrosion risk and provide relevant standards or references.
  18. Explain why humidity was set to 60% in the climate chamber tests in some cases, while 90% was used initially.
  19. Expand discussion on why higher humidity (90%) led to slower corrosion despite higher temperature.
  20. Discuss more explicitly the complex interaction of temperature, humidity, and chloride presence observed in your results.
  21. Provide a detailed explanation or hypothesis for why cracked beams showed lower corrosion rates.
  22. Figures (such as Figure 19 and Figure 21) need clearer labeling and consistent units across the axis.
  23. Provide a clearer interpretation or commentary on the variability observed in surface strain measurements.
  24. Discuss potential reasons for significant differences (approximately 3 kN) in the destructive test results for beams B8 and B9.
  25. Expand the discussion to include more clearly the potential long-term implications of your results for structural applications.
  26. Provide clearer references or literature that support your claims regarding oxygen availability and corrosion rate changes due to humidity variations.
  27. Caution that your conclusions are preliminary due to the small sample size; recommend performing tests on more specimens to generalize conclusions.
  28. Clearly indicate specific future research directions or potential variables worth exploring further.
  29. Figure annotations should be larger and clearer for improved readability.
  30. Ensure all tables clearly reference measurement points (P1, P2, P3 consistently), and units are uniformly presented.
  31. Perform a thorough language review for grammatical consistency and accuracy throughout the manuscript.
  32. The abstract could explicitly state the novelty or unique contribution of the study clearly.
  33. Strengthen conclusions by clearly stating how the outcomes impact practical engineering guidelines or recommendations.
  34. Clearly specify potential future studies, including exploring various environmental conditions or additional destructive tests.
Comments on the Quality of English Language

The English could be improved to more clearly express the research.

Author Response

Response to Reviewer 3 Comments

The manuscript addresses the corrosion behavior of prestressed concrete beams under varying temperature and humidity conditions, incorporating chloride ion migration and induced cracking conditions. The study presents valuable experimental data; however, several aspects require major revision to enhance clarity, strengthen the methodology, and improve the overall impact and rigor of the findings. Below are detailed recommendations for revision:

1) Clarify in the title if "pressed beams" refers specifically to prestressed concrete beams.

Ad 1) Yes, it's about prestressed concrete beams. The title has been changed.

2) Clearly state in the abstract the significance and novelty of your work compared to existing literature

Ad 2) To date, many studies can be found in the literature attempting to explain the effects of temperature and humidity on the rate of corrosion processes. However, it is difficult to analyze the results of these studies and draw unambiguous conclusions due to the different test conditions as well as different electrochemical test methods for corrosion rates.  Most of these studies concern concrete reinforced with ordinary steel. However, there is a lack of research and analysis conducted on prestressed elements.

The information has been added to the abstract.

3) Specify clearly how the "variable temperature and humidity conditions" differ from previous studies and their significance..

Ad 3) In the natural environment, there are usually both varying humidity and temperature conditions with the simultaneous effect of load on prestressed structures often causing scratching of the structure. Therefore, it is very important to comprehensively study such situations, considering the synergistic effects of environmental factors. Especially in an environment additionally exposed to the aggressive action of chloride ions. Most corrosion studies have been conducted on small test pieces or analyzed the impact of environmental factors independently. Therefore, there is a lack of research information needed to accurately analyze the progress of corrosion processes in elements of larger size and under the influence of different environmental conditions.

4) Consider adding "digital image correlation" and "chloride migration" to the keywords.

Ad 4) Thank you for the suggestion. I’ve added these keywords.

5) The description of stress corrosion cracking and hydrogen embrittlement is clear; however, briefly highlight the real-world implications or typical cases of failure.

Ad 5) Pitting corrosion contributes to a reduction in the cross-sectional area of the steel and a reduction in the load-bearing capacity of the structure. Stress corrosion, on the other hand, caused by an increase in the brittleness of steel, is considered more dominant than pitting corrosion. However, test results and theoretical considerations prefaced in some works indicate that this type of corrosion does not occur in the case of corrosion initiated by chloride ions.

It has also been shown that because of pitting corrosion, the proportion of brittle cracks in prestressing bands increases, but the ultimate tensile strength and deformation of steel bands significantly decrease because of the combined stress concentration and reduction of the cross-sectional area of steel bands. On the other hand, based on experience as well as field tests, it can be concluded that most of the damage and failure of prestressed structures is caused by pitting corrosion of steel. However, not enough research has been done to study the characteristics of pitting corrosion in prestressed structures, depending on the progress of corrosion processes. It is likely that the two causes not infrequently occur together in damaged prestressed structures, so regardless of the cause of rupture of steel strands, more studies of prestressed structures are needed to better understand these mechanisms and, as a result, prevent sudden structural failure.

6) Provide more references to recent studies on prestressed structures to contextualize the novelty of your research clearly.

Ad 6) I have added more references to recent studies on prestressed structures.

7) Explicitly articulate how your study addresses the knowledge gap identified in prestressed structures compared to previous work on ordinary reinforced concrete.

Ad 7) Whilst considerable research has been undertaken on steel corrosion in concrete, it is more focused on reinforcing steel than on prestressing steel. Although the underlying corrosion science is the same for both steels, the applicability of data on corrosion obtained from reinforcing steel to prestressing steel needs proof due to primarily the mechanical and physical differences between the two, in particular, the high level of stresses in prestressing steel strands and the microstructure of prestressing steel strands. There are fewer corrosion cells on the surface of prestressing strands than that of deformed bars. This reduced number of corrosion cells leads to faster growth of local corrosion at already corroded spots (cells). Since there is relatively more supply and less demand of water and oxygen for a small number of corrosion cells to sustain, pitting corrosion dominates in each strand (wire). Therefore, for corroded strands, more local and uneven corrosions occur.

From the analysis of the ultimate bond stress of all the specimens before, the presence of fatigue does improve the ultimate bond strength of the specimen, but this improvement is not infinite. When the fatigue accumulates to a certain extent, the ultimate adhesion of the specimen will be degraded.

Corrosion of the strand in a pre-tensioned prestressed concrete structure was found to degrade the tensile strength of the strand instead of its bond strength. The experimental investigation into the tensile behaviour of corroded strands presents an approximately linear degradation law for the tensile strength of strands with the corrosion rate.

8) Justify the choice of the w/c ratio (0.3) clearly in the context of corrosion studies.

Ad 8) It is known that the corrosion rate depends on many factors, including the w/c ratio of concrete. In this work, w/c = 0.3 was used because it is the most commonly used value in prestressed structures. The tests were carried out on elements made at Conbet's (Poznań, Poland) prestressed element factory, made for the purposes of this study, and this w/c value is maintained in the factory's product range. The only modification made especially for us was the use of one reinforcing string instead of two.

9 Provide additional details about the storage conditions of the beams prior to testing.

Ad 9) The beams immediately after fabrication at the factory were delivered to the Building of Laboratory of the Silesian University of Technology in Gliwice and stored in the conditions of the laboratory (humidity of about 30% and temperature of about 20° C).

10) Clarify why a 3% NaCl solution was chosen and its relevance to realistic scenarios

​ Ad 10) The authors of the paper concluded that research indicates that in a typical cell system with a volume of about 350 ml and a current voltage of 12 V, the concentration value of the source solution should not be less than 0.2 M, because below this concentration the sum of chlorides is not sufficient to ensure optimal efficiency of electric charge transport by chloride ions. The maximum value of the transfer number of chloride ions increased with an increase in the value of the concentration of the source solution used. This relationship was observed up to a concentration of 0.2 M of the source solution. Above the concentration of 0.2 M, the value of the chloride ion transfer number remained constant. It was decided to adopt a seawater concentration of about 3% (0.5M) close to that of a natural sea.

11) Clearly mention any challenges or limitations encountered in maintaining the electrical field uniformly during the migration tests.

Ad 11 There are two main issues with maintaining uniform electrical field during the migration tests. First, obtaining and maintaining the lowest possible resistance of the concrete, which is achieved by soaking the samples for at least 72 hours until the concrete is completely saturated with water. Then this moisture content is maintained by simultaneously immersing the test specimen in distilled water and filling the tank fixed from above with NaCl solution. The side walls of the test pieces are protected with a sealing agent. The change in the electric field can also be affected by the corrosion of the surface of the anode used in the test, so an electrode made of titanium coated with platinum is used in this test, which ensures the corrosion resistance of this electrode during the test.  The application of a small value of electric voltage guarantees the maintenance of a constant temperature during the test.

12) Explain the criteria for selecting measurement intervals (30-day cycles). The basics of the methodology of LPR should be explained by citing the reference: https://doi.org/10.1016/j.corsci.2023.111118

Ad 12 The interval of thirty days between measurements was determined based on migration tests of concrete samples taken from the tested elements and the determination of the time after which the concentration of chloride ions at the steel surface reaches the value at which corrosion can occur. Subsequent measurements were repeated at the same intervals.

The basics of the methodology of LPR was explained by citing the reference in the text.

13) Clarify how strain measurement accuracy was verified or calibrated.

Ad 13 Digital image correlation (DIC) measurement technique was used to examine the component's surface. Based on solid mechanics, the technique involves evaluating changes in geometry and the localisation of points before and after material deformation. The measurement relies on small rectangular areas called facets, which are relatively small, such as 15 x 15 pixels. Each facet has a unique pattern and overlaps with neighbouring facets within a range determined by the user (typically, 20-50% overlap is recommended). In this study, an overlap of 40% was chosen to capture local effects. Common areas were used to minimise errors in strain measurement, as each facet included elements from adjacent areas with the same boundary conditions. The size of the facets affects the accuracy and speed of calculations, with larger facets resulting in decreased measurement accuracy. The Aramis 6M system, consisting of two digital cameras, each with a resolution of 6 MPx, recording images in grey tones, was used in the tests. The measurements and processing of the results were carried out using GOM Correlate software, which was certified and calibrated for the camera configuration used. The measurement area of the Aramis 6M system ranges from 150x170 mm to 2150x2485 mm. An area with the following dimensions was used in the study:

- Initial test – area of 2150x2485 mm – 112 Px/cm2

- 3PBT and 4PBT – area of 1150x1340 mm – 390 Px/cm2

- Direct shear test – area of 150x170 mm – 23530 Px/cm2

The expanded measurement uncertainty should include calculations of measurement uncertainty due to inaccuracy of the device as well as the measurement itself. For the Aramis system, the expanded uncertainty can be calculated with 95% probability as:

U = ±(0.85 mm + 1.6 x 10-6 x L),

where: L - length of the target.

Below I attach the relevant data of the system manufacturer:

14) Include justification for selecting only two beams for destructive tests and discuss the implications on statistical reliability.

Ad 14

The measurements of the strength of the elements were preliminary. As can be seen from the results obtained, there should be more tests elements included: at least 3 and preferably 6 beams should be destroyed from the point of view of statistical analysis. However, due to the preliminary nature of the research, 10 beams from one tension track were obtained from the factory and we tried to use them to obtain as much information as possible. Of course, there are plans to study more elements.

15) Clearly explain why specific measurement points (P1, P2, P3) were selected.

Ad 15

Point P1 was chosen at a distance of 100 mm from the edge of the beam at a place where during storage they are supported and the stresses of steel from dead weight are close to 0. Similarly, point P3 was chosen at the same distance from the opposite end to average the values of measurement results obtained. Point P2 was chosen in the middle of the beam span at a place where the prestressing steel is most stressed. In summary, it was decided to choose points P1 and P2 because of the importance of obtaining corrosion current measurements at the most and least stressed point of the prestressing steel. Point P3 was chosen to average the measurements.

16) Figures showing polarization curves should include clearer axis labeling and scales for enhanced readability.

Ad 16 The drawings have been corrected.

17) Clarify the reason behind classifying corrosion current density below 5 µA as "low" corrosion risk and provide relevant standards or references.

Ad 17 These criteria were adopted from publications [28], [30], [31] and Gekor's instructions.

18) Explain why humidity was set to 60% in the climate chamber tests in some cases, while 90% was used initially.

Ad 18 Each time the humidity in the chamber was 90%. It was mistakenly described as 60% humidity.

19) Expand discussion on why higher humidity (90%) led to slower corrosion despite higher temperature

Ad 19 If we compare the corrosion rate in two media with high moisture content or immersed directly in a solution, we observe an increase in the corrosion rate with increasing temperature. On the other hand, in semi-dry or low-moisture concrete, we observe a decrease in corrosion processes with an increase in temperature. If we superimpose both factors, we find that with very high moisture content or total immersion, corrosion processes occur more slowly than in semi-dry or low-moisture samples. The decisive aspect of the corrosion process here is the access of oxygen, which is easier in samples with lower humidity than in samples with high humidity.

20) Discuss more explicitly the complex interaction of temperature, humidity, and chloride presence observed in your results.

Ad 20 Storing the beam in high humidity conditions extinguished the corrosion processes previously initiated by the presence of chloride ions. Therefore, it can be assumed that the decisive factor affecting the increase in the rate of these processes is the access of oxygen, hindered in this case by the saturation of concrete with moisture.

21) Provide a detailed explanation or hypothesis for why cracked beams showed lower corrosion rates.

Ad 21 The higher values of corrosion current observed in uncracked beams compared to cracked beams and beams with additional loading occurring both under laboratory conditions and in the climatic chamber can be attributed here to changes in the corrosion rate caused by the effect of stresses acting on the beam. Since the reinforcing string is in tension in the prestressed concrete member, we can expect corrosion processes to be faster in it than in ordinary reinforcement. On the other hand, once scratching is achieved in a prestressed concrete element, the prestressing forces are reduced, which contributes to a decrease in the speed of corrosion processes.      

22) Figures (such as Figure 19 and Figure 21) need clearer labeling and consistent units across the axis

Ad 22 Figures 19 and 21 have been corrected

23) Provide a clearer interpretation or commentary on the variability observed in surface strain measurements.

Ad 23 As shown in the [ 39] paper, studies of the elastic modulus of ordinary concrete indicate a slight reduction in Young's modulus associated with the addition of chlorides directly to the concrete mixture. In contrast, a definite increase of about 42% in the value of the modulus of elasticity was obtained in lightweight concrete in samples treated with chloride ions in an electric field.

24) Discuss potential reasons for significant differences (approximately 3 kN) in the destructive test results for beams B8 and B9.

Ad 24 The occurrence of such a large difference in the results of strength tests is difficult to explain after performing only two tests, because we can not predict whether some value should not be rejected for various reasons: whether due to inaccuracy of measurement or insufficient quality of the selected component. It is necessary to perform more tests.

25) Expand the discussion to include more clearly the potential long-term implications of your results for structural applications.

Ad 25 The results obtained from tests carried out under complex loading conditions of both variable temperature, humidity and long-term loading will allow more accurate prediction of the performance of the structure under natural operating conditions and predict possible risks caused by corrosion of steel. Numerical modeling of the structure's behavior will make it possible to predict possible threats occurring under conditions of real climate and aggressive environment containing chloride ions. This will help avoid problems at the stage of designing prestressed structures.

26) Provide clearer references or literature that support your claims regarding oxygen availability and corrosion rate changes due to humidity variations.

Ad 26 Supplemented in the text.

27) Caution that your conclusions are preliminary due to the small sample size; recommend performing tests on more specimens to generalize conclusions.

Ad 27 It is also necessary to carry out more tests to confirm the preliminary test results obtained.

28) Clearly indicate specific future research directions or potential variables worth exploring further.

Ad 28 Further tests should focus on:

1) determining close correlations between adhesion and concentration of Cl– ions,

2) specifying changes in the area of tendons caused by pitting corrosion,

3) detailed description of relationships for proper determining losses in prestressing, which are

caused by bond strength loss, changes in elasticity modulus of concrete, and the formation of

corrosion products.

29) Figure annotations should be larger and clearer for improved readability.

Ad 29 The descriptions of the drawings have been enlarged.

30) Ensure all tables clearly reference measurement points (P1, P2, P3 consistently), and units are uniformly presented.

Ad 30 The tables have been checked and corrected.

31) Perform a thorough language review for grammatical consistency and accuracy throughout the manuscript.

Ad 31 The English language has been corrected.

32) The abstract could explicitly state the novelty or unique contribution of the study clearly.

Ad 32 Abstract completed.

33) Strengthen conclusions by clearly stating how the outcomes impact practical engineering guidelines or recommendations.

Ad 33 The research is at a preliminary stage. After more research is conducted, it will be possible to make some recommendations on the design of prestressed structures.

34) Clearly specify potential future studies, including exploring various environmental conditions or additional destructive tests.

Ad 34 The following tests are planned for the next research stages

  1. a) strength tests of non-corroded beams (additionally minimum 4 beams)
  2. b) strength tests of beams after the next stage of corrosion processes under changed temperature conditions
  3. c) tests of corrosion development under low temperature conditions
  4. d) strength tests of corroded and non-corroded reinforcement strings
  5. e) testing of corrosion loss by gravimetric method and using optical microscope of rebar strings
  6. f) testing the effect of low temperatures on the development of corrosion processes
  7. g) testing under conditions of concrete carbonation
  8. h) modeling of corrosion processes

Reviewer 4 Report

Comments and Suggestions for Authors

1. Fig 1. Mark the location of the tendon in the sample.
2. Please clarify the direction of compressive strength measured.
3. Clarify the meaning of 'w/c ratio'.
4. Clarify the heat treatment of 'prestressing steel Y2060-S7' and the phase/microstructure present.
5. 'Immediately before the test, the beams were soaked in tap water to improve the concreteʹs resistivity': please clarify the composition of tap water. Does it contain chlorine or other tracer elements? Will a different composition of tap water influence the measurements?
6. Corrosion test: are the electrodes in contact with the tendon? Please clarify this point in the manuscript.
7. Clarify the meaning of EIS method.
8. The results presented are comprehensive. However, more discussions are needed to explain observed results.
9. Conclusion part: there is no point using B1/B2/B3/B4 notation without explaining what each beam stands for. It is similar to the previous comment that the results need to be more discussed and concluded. 
10. It would be better to show typical microstructure for samples after the corrosion test. 

Author Response

Response to Reviewer 4 Comments

1) Fig 1. Mark the location of the tendon in the sample.

Ad 1) I’ve added the marking on the figure.

2) Please clarify the direction of compressive strength measured.

Ad 2) I’ve explained the load P action diagram in Fig. 7b

3) Clarify the meaning of 'w/c ratio'.

Ad 3) Water/cement ratio (w/c) is the ratio of the effective water content to the cement content in a concrete mix.

4) Clarify the heat treatment of 'prestressing steel Y2060-S7' and the phase/microstructure present.

Ad 4)  The production process of strands proceeds as follows: material for wire rod is smelted in converters. The resulting steel has a carbon content of about 1%. Slabs for hot rolling are produced using continuous casting technology. Hot rolling results in a bar with a diameter of 5-8 mm, which is the starting material for drawing. A controlled cooling process in air after rolling makes it possible to obtain the required structure inside the material. Due to further cold processing, a bainitic structure throughout is optimal, as it has fine and dispersed cementite. The lack of cementite makes the microstructure more resistant to brittle fracture. Hot-rolled bar has a layer of scale consisting of iron oxides on its surface (FeO, Fe3O4,Fe2O3), which prevents the drawing process, as these oxides are hard and brittle. For this reason, they are removed chemically during etching in hydrochloric or sulfuric acid with an inhibitor. Then, layers of manganese, iron or zinc phosphates are applied to the wire rod with a clean metallic surface in suitable baths, which allow the drawing speed and size of individual nits to be increased. The phosphate layer also partially protects the surface from corrosion. As a result of the drawing process, grains having a random orientation deform longitudinally and become oriented according to the direction of stretching. Under the influence of the deformation, the material strengthens, resulting in an increase in strength properties while plastic properties decrease. A single-sided weave is produced from the wires. The final step is thermomechanical processing, which involves pulling through tensile units and passing the strand through a high-frequency induction furnace.

5) 'Immediately before the test, the beams were soaked in tap water to improve the concreteʹs resistivity': please clarify the composition of tap water. Does it contain chlorine or other tracer elements? Will a different composition of tap water influence the measurements?

Ad 5) Each liter of tap water contains 0.5 g of minerals, including: 0.01 g of magnesium and 0.07 g of calcium. The chloride content in drinking water should not exceed 0.0003 g/l. The exemplary concentration of minerals in concrete pore liquid is Ca - 0.19 g/l, Na - 2.1 g/l, K - 14.0 g/l. The use of tap water in the case of impregnation of concrete elements does not affect the resistance of concrete and the obtained measurement results. Moreover, the elements were manufactured using tap water as is standard practice in the industry.

6) Corrosion test: are the electrodes in contact with the tendon? Please clarify this point in the manuscript.

Ad 6) I’ve added the explanation in the text.

7) Clarify the meaning of EIS method.

Ad 7) I mistakenly included the EIS method in the text, which was not used in these studies. EIS Electrochemical Impedance Spectroscopy means one of the non-destructive methods of measuring corrosion current.

8) The results presented are comprehensive. However, more discussions are needed to explain observed results.

Ad 8) I have supplemented the discussion.

9 Conclusion part: there is no point using B1/B2/B3/B4 notation without explaining what each beam stands for. It is similar to the previous comment that the results need to be more discussed and concluded.

Ad 9) I have supplemented the discussion and conclusions sections.

10) It would be better to show typical microstructure for samples after the corrosion test.

​ Ad 10) The tests of the beams are continued after their destruction in the strength tests; tests of the microstructure of the samples are planned and these tests will be supplemented in subsequent works.

Round 2

Reviewer 1 Report

Comments and Suggestions for Authors

Authors have revised the manuscript, it can be published in the present form.

Reviewer 3 Report

Comments and Suggestions for Authors

The authors successfully answered the comments and the revised version can be considered for publication.

Reviewer 4 Report

Comments and Suggestions for Authors

The authors addressed most of my questions.